# Neural Temporal-Difference Learning Converges to Global Optima

Qi Cai [*]    Zhuoran Yang [†]    Jason D. Lee [‡]    Zhaoran Wang [*]

## Abstract

Temporal-difference learning (TD), coupled with neural networks, is among the most fundamental building blocks of deep reinforcement learning. However, due to the nonlinearity in value function approximation, such a coupling leads to non-convexity and even divergence in optimization. As a result, the global convergence of neural TD remains unclear. In this paper, we prove for the first time that neural TD converges at a sublinear rate to the global optimum of the mean-squared projected Bellman error for policy evaluation. In particular, we show how such global convergence is enabled by the overparametrization of neural networks, which also plays a vital role in the empirical success of neural TD.[1]

## 1 Introduction

Given a policy, temporal-different learning (TD) [49] aims to learn the corresponding (action-)value function by following the semigradients of the mean-squared Bellman error in an online manner. As the most-used policy evaluation algorithm, TD serves as the "critic" component of many reinforcement learning algorithms, such as the actor-critic algorithm [31] and trust-region policy optimization [47]. In particular, in deep reinforcement learning, TD is often applied to learn value functions parametrized by neural networks [36, 39, 24], which gives rise to neural TD. As policy improvement relies crucially on policy evaluation, the optimization efficiency and statistical accuracy of neural TD are critical to the performance of deep reinforcement learning. Towards theoretically understanding deep reinforcement learning, the goal of this paper is to characterize the convergence of neural TD.

Despite the broad applications of neural TD, its convergence remains rarely understood. Even with linear value function approximation, the nonasymptotic convergence of TD remains open until recently [6, 33, 14, 48, 45], although its asymptotic convergence is well understood [28, 55, 9, 32, 8]. Meanwhile, with nonlinear value function approximation, TD is known to diverge in general [4, 11, 55]. To remedy this issue, [7] propose nonlinear (gradient) TD, which uses the tangent vectors of nonlinear value functions in place of the feature vectors in linear TD. Unlike linear TD, which converges to the global optimum of the mean-squared projected Bellman error (MSPBE), nonlinear TD is only guaranteed to converge to a local optimum asymptotically. As a result, the statistical accuracy of the value function learned by nonlinear TD remains unclear. In contrast to such conservative theory, neural TD, which straightforwardly combines TD with neural networks without the explicit local linearization in nonlinear TD, often learns a desired value function that generalizes well to unseen states in practice [18, 2, 26]. Hence, a gap separates theory from practice.

---

[*]Department of Industrial Engineering and Management Sciences, Northwestern University
[†]Department of Operations Research and Financial Engineering, Princeton University
[‡]Department of Electronic Engineering, Princeton University

[1]Beyond policy evaluation, we establish the global convergence of neural (soft) Q-learning, which is further connected to that of policy gradient algorithms. See https://arxiv.org/abs/1905.10027 for the full version.

There exist three obstacles towards closing such a theory-practice gap: (i) MSPBE has an expectation over the transition dynamics within the squared loss, which forbids the construction of unbiased stochastic gradients [50]. As a result, even with linear value function approximation, TD largely eludes the classical optimization framework, as it follows biased stochastic semigradients. (ii) When the value function is parametrized by a neural network, MSPBE is nonconvex in the weights of the neural network, which may introduce undesired stationary points such as local optima and saddle points [30]. As a result, even an ideal algorithm that follows the population gradients of MSPBE may get trapped. (iii) Due to the interplay between the bias in stochastic semigradients and the nonlinearity in value function approximation, neural TD may even diverge [4, 11, 55], instead of converging to an undesired stationary point, as it lacks the explicit local linearization in nonlinear TD [7]. Such divergence is also not captured by the classical optimization framework.

**Contribution.** Towards bridging theory and practice, we establish the first nonasymptotic global rate of convergence of neural TD. In detail, we prove that randomly initialized neural TD converges to the global optimum of MSPBE at the rate of $1/T$ with population semigradients and at the rate of $1/\sqrt{T}$ with stochastic semigradients. Here $T$ is the number of iterations and the (action-)value function is parametrized by a sufficiently wide two-layer neural network. Moreover, we prove that the projection in MSPBE allows for a sufficiently rich class of functions, which has the same representation power of a reproducing kernel Hilbert space associated with the random initialization. As a result, for a broad class of reinforcement learning problems, neural TD attains zero MSPBE.

At the core of our analysis is the overparametrization of the two-layer neural network for value function approximation [59, 41, 1, 3], which enables us to circumvent the three obstacles above. In particular, overparametrization leads to an implicit local linearization that varies smoothly along the solution path, which mirrors the explicit one in nonlinear TD [7]. Such an implicit local linearization enables us to circumvent the third obstacle of possible divergence. Moreover, overparametrization allows us to establish a notion of one-point monotonicity [25, 19] for the semigradients followed by neural TD, which ensures its evolution towards the global optimum of MSPBE along the solution path. Such a notion of monotonicity enables us to circumvent the first and second obstacles of bias and nonconvexity. Broadly speaking, our theory backs the empirical success of overparametrized neural networks in deep reinforcement learning. In particular, we show that instead of being a curse, overparametrization is indeed a blessing for minimizing MSPBE in the presence of bias, nonconvexity, and even divergence.

**More Related Work.** There is a large body of literature on the convergence of linear TD under both asymptotic [28, 55, 9, 32, 8] and nonasymptotic [6, 33, 14, 48] regimes. See [16] for a detailed survey. In particular, our analysis is based on the recent breakthrough in the nonasymptotic analysis of linear TD [6] and its extension to linear Q-learning [60]. An essential step of our analysis is bridging the evolution of linear TD and neural TD through the implicit local linearization induced by overparametrization.

To incorporate nonlinear value function approximation into TD, [7] propose the first convergent nonlinear TD based on explicit local linearization, which however only converges to a local optimum of MSPBE. See [21, 5] for a detailed survey. In contrast, we prove that, with the implicit local linearization induced by overparametrization, neural TD, which is simpler to implement and more widely used in deep reinforcement learning than nonlinear TD, provably converges to the global optimum of MSPBE.

There exist various extensions of TD, including least-squares TD [12, 10, 34, 22, 56] and gradient TD [51, 52, 7, 37, 17, 57, 54]. In detail, least-squares TD is based on batch update, which loses the computational and statistical efficiency of the online update in TD. Meanwhile, gradient TD follows unbiased stochastic gradients, but at the cost of introducing another optimization variable. Such a reformulation leads to bilevel optimization, which is less stable in practice when combined with neural networks [42]. As a result, both extensions of TD are less widely used in deep reinforcement learning [18, 2, 26]. Moreover, when using neural networks for value function approximation, the convergence to the global optimum of MSPBE remains unclear for both extensions of TD.

Our work is also related to the recent breakthrough in understanding overparametrized neural networks, especially their generalization error [59, 41, 1, 3]. See [20] for a detailed survey. In particular, [15, 1, 3, 13, 29, 35] characterize the implicit local linearization in the context of supervised learning, where we train an overparametrized neural network by following the stochastic gradients

of the mean-squared error. In contrast, neural TD does not follow the stochastic gradients of any objective function, hence leading to possible divergence, which makes the convergence analysis more challenging.

## 2 Background

In Section 2.1, we briefly review policy evaluation in reinforcement learning. In Section 2.2, we introduce the corresponding optimization formulations.

### 2.1 Policy Evaluation

We consider a Markov decision process $(\mathcal{S}, \mathcal{A}, \mathcal{P}, r, \gamma)$, in which an agent interacts with the environment to learn the optimal policy that maximizes the expected total reward. At the $t$-th time step, the agent has a state $s_t \in \mathcal{S}$ and takes an action $a_t \in \mathcal{A}$. Upon taking the action, the agent enters the next state $s_{t+1} \in \mathcal{S}$ according to the transition probability $\mathcal{P}(\cdot \,|\, s_t, a_t)$ and receives a random reward $r_t = r(s_t, a_t)$ from the environment. The action that the agent takes at each state is decided by a policy $\pi : \mathcal{S} \to \Delta$, where $\Delta$ is the set of all probability distributions over $\mathcal{A}$. The performance of policy $\pi$ is measured by the expected total reward, $J(\pi) = \mathbb{E}[\sum_{t=0}^{\infty} \gamma^t r_t \,|\, a_t \sim \pi(s_t)]$, where $\gamma < 1$ is the discount factor.

Given policy $\pi$, policy evaluation aims to learn the following two functions, the value function $V^\pi(s) = \mathbb{E}[\sum_{t=0}^{\infty} \gamma^t r_t \,|\, s_0 = s, a_t \sim \pi(s_t)]$ and the action-value function (Q-function) $Q^\pi(s, a) = \mathbb{E}[\sum_{t=0}^{\infty} \gamma^t r_t \,|\, s_0 = s, a_0 = a, a_t \sim \pi(s_t)]$. Both functions form the basis for policy improvement. Without loss of generality, we focus on learning the Q-function in this paper. We define the Bellman evaluation operator,

$$\mathcal{T}^\pi Q(s, a) = \mathbb{E}[r(s, a) + \gamma Q(s', a') \,|\, s' \sim \mathcal{P}(\cdot \,|\, s, a), a' \sim \pi(s')], \tag{2.1}$$

for which $Q^\pi$ is the fixed point, that is, the solution to the Bellman equation $Q = \mathcal{T}^\pi Q$.

### 2.2 Optimization Formulation

Corresponding to (2.1), we aim to learn $Q^\pi$ by minimizing the mean-squared Bellman error (MSBE),

$$\min_{\theta} \mathrm{MSBE}(\theta) = \mathbb{E}_{(s,a)\sim\mu}\big[\big(\widehat{Q}_\theta(s, a) - \mathcal{T}^\pi \widehat{Q}_\theta(s, a)\big)^2\big], \tag{2.2}$$

where the Q-function is parametrized as $\widehat{Q}_\theta$ with parameter $\theta$. Here $\mu$ is the stationary distribution of $(s, a)$ corresponding to policy $\pi$. Due to Q-function approximation, we focus on minimizing the following surrogate of MSBE, namely the projected mean-squared Bellman error (MSPBE),

$$\min_{\theta} \mathrm{MSPBE}(\theta) = \mathbb{E}_{(s,a)\sim\mu}\big[\big(\widehat{Q}_\theta(s, a) - \Pi_{\mathcal{F}} \mathcal{T}^\pi \widehat{Q}_\theta(s, a)\big)^2\big]. \tag{2.3}$$

Here $\Pi_{\mathcal{F}}$ is the projection onto a function class $\mathcal{F}$. For example, for linear Q-function approximation [49], $\mathcal{F}$ takes the form $\{\widehat{Q}_{\theta'} : \theta' \in \Theta\}$, where $\widehat{Q}_{\theta'}$ is linear in $\theta'$ and $\Theta$ is the set of feasible parameters. As another example, for nonlinear Q-function approximation [7], $\mathcal{F}$ takes the form $\{\widehat{Q}_\theta + \nabla_\theta \widehat{Q}_\theta^\top (\theta' - \theta) : \theta' \in \Theta\}$, which consists of the local linearization of $\widehat{Q}_{\theta'}$ at $\theta$.

Throughout this paper, we assume that we are able to sample tuples in the form of $(s, a, r, s', a')$ from the stationary distribution of policy $\pi$ in an independent and identically distributed manner, although our analysis can be extended to handle temporal dependence using the proof techniques of [6]. With a slight abuse of notation, we use $\mu$ to denote the stationary distribution of $(s, a, r, s', a')$ corresponding to policy $\pi$ and any of its marginal distributions.

## 3 Neural Temporal-Difference Learning

TD updates the parameter $\theta$ of the Q-function by taking the stochastic semigradient descent step [49, 53, 50],

$$\theta' \leftarrow \theta - \eta \cdot \big(\widehat{Q}_\theta(s, a) - r(s, a) - \gamma \widehat{Q}_\theta(s', a')\big) \cdot \nabla_\theta \widehat{Q}_\theta(s, a), \tag{3.1}$$

which corresponds to the MSBE in (2.2). Here $(s, a, r, s', a') \sim \mu$ and $\eta > 0$ is the stepsize. In a more general context, (3.1) is referred to as TD(0). In this paper, we focus on TD(0), which is abbreviated as TD, and leave the extension to TD($\lambda$) to future work.

In the sequel, we denote the state-action pair $(s, a) \in \mathcal{S} \times \mathcal{A}$ by a vector $x \in \mathcal{X} \subseteq \mathbb{R}^d$ with $d > 2$. We consider $\mathcal{S}$ to be continuous and $\mathcal{A}$ to be finite. Without loss of generality, we assume that $\|x\|_2 = 1$ and $|r(x)|$ is upper bounded by a constant $\overline{r}$ for any $x \in \mathcal{X}$. We use a two-layer neural network

$$\widehat{Q}(x; W) = \frac{1}{\sqrt{m}} \sum_{r=1}^{m} b_r \sigma(W_r^\top x) \tag{3.2}$$

to parametrize the Q-function. Here $\sigma$ is the rectified linear unit (ReLU) activation function $\sigma(y) = \max\{0, y\}$ and the parameter $\theta = (b_1, \ldots, b_m, W_1, \ldots, W_m)$ are initialized as $b_r \sim \text{Unif}(\{-1, 1\})$ and $W_r \sim N(0, I_d/d)$ for any $r \in [m]$ independently. During training, we only update $W = (W_1, \ldots, W_m) \in \mathbb{R}^{md}$, while keeping $b = (b_1, \ldots, b_m) \in \mathbb{R}^m$ fixed as the random initialization. To ensure global convergence, we incorporate an additional projection step with respect to $W$. See Algorithm 1 for a detailed description.

---

**Algorithm 1** Neural TD

1: **Initialization:** $b_r \sim \text{Unif}(\{-1, 1\})$, $W_r(0) \sim N(0, I_d/d)$ $(r \in [m])$, $\overline{W} = W(0)$,
           $S_B = \{W \in \mathbb{R}^{md} : \|W - W(0)\|_2 \le B\}$ $(B > 0)$
2: **For** $t = 0$ to $T - 2$:
3:      Sample a tuple $(s, a, r, s', a')$ from the stationary distribution $\mu$ of policy $\pi$
4:      Let $x = (s, a)$, $x' = (s', a')$
5:      Bellman residual calculation: $\delta \leftarrow \widehat{Q}(x; W(t)) - r - \gamma \widehat{Q}(x'; W(t))$
6:      TD update: $\widetilde{W}(t+1) \leftarrow W(t) - \eta \delta \cdot \nabla_W \widehat{Q}(x; W(t))$
7:      Projection: $W(t+1) \leftarrow \text{argmin}_{W \in S_B} \|W - \widetilde{W}(t+1)\|_2$
8:      Averaging: $\overline{W} \leftarrow \frac{t+1}{t+2} \cdot \overline{W} + \frac{1}{t+2} \cdot W(t+1)$
9: **End For**
10: **Output:** $\widehat{Q}_{\text{out}}(\cdot) \leftarrow \widehat{Q}(\cdot; \overline{W})$

---

To understand the intuition behind the global convergence of neural TD, note that for the TD update in (3.1), we have from (2.1) that

$$\mathbb{E}_{(s,a,r,s',a') \sim \mu}\big[\big(\widehat{Q}_\theta(s, a) - r(s, a) - \gamma \widehat{Q}_\theta(s', a')\big) \cdot \nabla_\theta \widehat{Q}_\theta(s, a)\big]$$

$$= \mathbb{E}_{(s,a) \sim \mu}\big[\big(\widehat{Q}_\theta(s, a) - \mathbb{E}[r(s, a) + \gamma Q(s', a') \,|\, s' \sim \mathcal{P}(\cdot \,|\, s, a), a' \sim \pi(s')]\big) \cdot \nabla_\theta \widehat{Q}_\theta(s, a)\big]$$

$$= \mathbb{E}_{(s,a) \sim \mu}\big[\underbrace{\big(\widehat{Q}_\theta(s, a) - \mathcal{T}^\pi \widehat{Q}_\theta(s, a)\big)}_{\text{(i)}} \cdot \underbrace{\nabla_\theta \widehat{Q}_\theta(s, a)}_{\text{(ii)}}\big]. \tag{3.3}$$

Here (i) is the Bellman residual at $(s, a)$, while (ii) is the gradient of the first term in (i). Although the TD update in (3.1) resembles the stochastic gradient descent step for minimizing a mean-squared error, it is not an unbiased stochastic gradient of any objective function. However, we show that the TD update yields a descent direction towards the global optimum of the MSPBE in (2.3). Moreover, as the neural network becomes wider, the function class $\mathcal{F}$ that $\Pi_\mathcal{F}$ projects onto in (2.3) becomes richer. Correspondingly, the MSPBE reduces to the MSBE in (2.2) as the projection becomes closer to identity, which implies the recovery of the desired Q-function $Q^\pi$ such that $Q^\pi = \mathcal{T}^\pi Q^\pi$. See Section 4 for a more rigorous characterization.

## 4 Main Results

In Section 4.1, we characterize the global optimality of the stationary point attained by Algorithm 1 in terms of minimizing the MSPBE in (2.3) and its other properties. In Section 4.2, we establish the nonasymptotic global rates of convergence of neural TD to the global optimum of the MSPBE when following the population semigradients in (3.3) and the stochastic semigradients in (3.1), respectively.

We use the subscript $\mathbb{E}_\mu[\cdot]$ to denote the expectation over the randomness of the tuple $(s, a, r, s, a')$ (or its concise form $(x, r, x')$) conditional on all other randomness, e.g., the random initialization

and the random current iterate. Meanwhile, we use the subscript $\mathbb{E}_{\text{init},\mu}[\cdot]$ when we are taking the expectation over all randomness, including the random initialization.

## 4.1 Properties of Stationary Point

We consider the population version of the TD update in Line 6 of Algorithm 1,

$$\widetilde{W}(t+1) \leftarrow W(t) - \eta \cdot \mathbb{E}_\mu \big[ \delta \big( x, r, x'; W(t) \big) \cdot \nabla_W \widehat{Q}\big( x; W(t) \big) \big], \tag{4.1}$$

where $\mu$ is the stationary distribution and $\delta(x, r, x'; W(t)) = \widehat{Q}(x; W(t)) - r - \gamma \widehat{Q}(x'; W(t))$ is the Bellman residual at $(x, r, x')$. The stationary point $W^\dagger$ of (4.1) satisfies the following stationarity condition,

$$\mathbb{E}_\mu [\delta(x, r, x'; W^\dagger) \cdot \nabla_W \widehat{Q}(x; W^\dagger)]^\top (W - W^\dagger) \geq 0, \quad \text{for any } W \in S_B. \tag{4.2}$$

Also, note that

$$\widehat{Q}(x; W) = \frac{1}{\sqrt{m}} \sum_{r=1}^m b_r \sigma(W_r^\top x) = \frac{1}{\sqrt{m}} \sum_{r=1}^m b_r \, \mathbb{1}\{W_r^\top x > 0\} W_r^\top x$$

and $\nabla_{W_r} \widehat{Q}(x; W) = b_r \, \mathbb{1}\{W_r^\top x > 0\} x$ almost everywhere in $\mathbb{R}^{md}$. Meanwhile, recall that $S_B = \{W \in \mathbb{R}^{md} : \|W - W(0)\|_2 \leq B\}$. We define the function class

$$\mathcal{F}_{B,m}^\dagger = \left\{ \frac{1}{\sqrt{m}} \sum_{r=1}^m b_r \, \mathbb{1}\{(W_r^\dagger)^\top x > 0\} W_r^\top x : W \in S_B \right\}, \tag{4.3}$$

which consists of the local linearization of $\widehat{Q}(x; W)$ at $W = W^\dagger$. Then (4.2) takes the following equivalent form

$$\big\langle \widehat{Q}(\cdot; W^\dagger) - \mathcal{T}^\pi \widehat{Q}(\cdot; W^\dagger), f(\cdot) - \widehat{Q}(\cdot; W^\dagger) \big\rangle_\mu \geq 0, \quad \text{for any } f \in \mathcal{F}_{B,m}^\dagger, \tag{4.4}$$

which implies $\widehat{Q}(\cdot; W^\dagger) = \Pi_{\mathcal{F}_{B,m}^\dagger} \mathcal{T}^\pi \widehat{Q}(\cdot; W^\dagger)$ by the definition of the projection induced by $\langle \cdot, \cdot \rangle_\mu$. By (2.3), $\widehat{Q}(\cdot; W^\dagger)$ is the global optimum of the MSPBE that corresponds to the projection onto $\mathcal{F}_{B,m}^\dagger$.

Intuitively, when using an overparametrized neural network with width $m \to \infty$, the average variation in each $W_r$ diminishes to zero. Hence, roughly speaking, we have $\mathbb{1}\{W_r(t)^\top x > 0\} = \mathbb{1}\{W_r(0)^\top x > 0\}$ with high probability for any $t \in [T]$. As a result, the function class $\mathcal{F}_{B,m}^\dagger$ defined in (4.3) approximates

$$\mathcal{F}_{B,m} = \left\{ \frac{1}{\sqrt{m}} \sum_{r=1}^m b_r \, \mathbb{1}\{W_r(0)^\top x > 0\} W_r^\top x : W \in S_B \right\}. \tag{4.5}$$

In the sequel, we show that, to characterize the global convergence of Algorithm 1 with a sufficiently large $m$, it suffices to consider $\mathcal{F}_{B,m}$ in place of $\mathcal{F}_{B,m}^\dagger$, which simplifies the analysis, since the distribution of $W(0)$ is given. To this end, we define the approximate stationary point $W^*$ with respect to the function class $\mathcal{F}_{B,m}$ defined in (4.5).

**Definition 4.1** (Approximate Stationary Point $W^*$). If $W^* = (W_1^*, \ldots, W_m^*) \in S_B$ satisfies

$$\mathbb{E}_\mu [\delta_0(x, r, x'; W^*) \cdot \nabla_W \widehat{Q}_0(x; W^*)]^\top (W - W^*) \geq 0, \quad \text{for any } W \in S_B, \tag{4.6}$$

where we define

$$\widehat{Q}_0(x; W) = \frac{1}{\sqrt{m}} \sum_{r=1}^m b_r \, \mathbb{1}\{W_r(0)^\top x > 0\} W_r^\top x, \tag{4.7}$$

$$\delta_0(x, r, x'; W) = \widehat{Q}_0(x; W) - r - \gamma \widehat{Q}_0(x'; W), \tag{4.8}$$

then we say that $W^*$ is an approximate stationary point of the population update in (4.1). Here $W^*$ depends on the random initialization $b = (b_1, \ldots, b_m)$ and $W(0) = (W_1(0), \ldots, W_m(0))$.

The next lemma proves that such an approximate stationary point always exists, since it corresponds to the fixed point of the operator $\Pi_{\mathcal{F}_{B,m}}\mathcal{T}^\pi$, which is a contraction in the $\ell_2$-norm associated with the stationary distribution $\mu$.

**Lemma 4.2** (Existence and Optimality of $W^*$)**.** There exists an approximate stationary point $W^*$ for any $b \in \{-1, 1\}^m$ and $W(0) \in \mathbb{R}^{md}$. Also, $\widehat{Q}_0(\cdot\,; W^*)$ is the global optimum of the MSPBE that corresponds to the projection onto $\mathcal{F}_{B,m}$ in (4.5).

*Proof.* See Appendix B.1 for a detailed proof. $\quad\square$

## 4.2 Global Convergence

In this section, we establish the main results on the global convergence of neural TD in Algorithm 1. We first lay out the following regularity condition on the stationary distribution $\mu$.

**Assumption 4.3** (Regularity of Stationary Distribution $\mu$)**.** There exists a constant $c_0 > 0$ such that for any $\tau \geq 0$ and $w \sim N(0, I_d/d)$, it holds almost surely that

$$\mathbb{E}_\mu\big[\mathbb{1}\{|w^\top x| \leq \tau\} \,\big|\, w\big] \leq c_0 \cdot \tau/\|w\|_2. \tag{4.9}$$

Assumption 4.3 regularizes the density of $\mu$ in terms of the marginal distribution of $x$. In particular, it is straightforwardly implied when the density of $\mu$ in terms of state $s$ is upper bounded.

**Population Update:** The next theorem establishes the nonasymptotic global rate of convergence of neural TD when it follows population semigradients. Recall that the approximate stationary point $W^*$ and $\widehat{Q}_0(\cdot\,; W^*)$ are defined in Definition 4.1. Also, $B$ is the radius of the set of feasible $W$, which is defined in Algorithm 1, $T$ is the number of iterations, $\gamma$ is the discount factor, and $m$ is the width of the neural network in (3.2).

**Theorem 4.4** (Convergence of Population Update)**.** We set $\eta = (1-\gamma)/8$ in Algorithm 1 and replace the TD update in Line 6 by the population update in (4.1). Under Assumption 4.3, the output $\widehat{Q}_{\text{out}}$ of Algorithm 1 satisfies

$$\mathbb{E}_{\text{init},\mu}\big[\big(\widehat{Q}_{\text{out}}(x) - \widehat{Q}_0(x; W^*)\big)^2\big] \leq \frac{16B^2}{(1-\gamma)^2 T} + O(B^3 m^{-1/2} + B^{5/2} m^{-1/4}),$$

where the expectation is taken with respect to all randomness, including the random initialization and the stationary distribution $\mu$.

*Proof.* The key to the proof of Theorem 4.4 is the one-point monotonicity of the population semigradient $\overline{g}(t)$, which is established through the local linearization $\widehat{Q}_0(x; W)$ of $\widehat{Q}(x; W)$. See Appendix C.5 for a detailed proof. $\quad\square$

**Stochastic Update:** To further prove the global convergence of neural TD when it follows stochastic semigradients, we first establish an upper bound of their variance, which affects the choice of the stepsize $\eta$. For notational simplicity, we define the stochastic and population semigradients as

$$g(t) = \delta\big(x, r, x'; W(t)\big) \cdot \nabla_W \widehat{Q}\big(x; W(t)\big), \quad \overline{g}(t) = \mathbb{E}_\mu[g(t)]. \tag{4.10}$$

**Lemma 4.5** (Variance Bound)**.** There exists $\sigma_g^2 = O(B^2)$ such that the variance of the stochastic semigradient is upper bounded as $\mathbb{E}_{\text{init},\mu}[\|g(t) - \overline{g}(t)\|_2^2] \leq \sigma_g^2$ for any $t \in [T]$.

*Proof.* See Appendix B.2 for a detailed proof. $\quad\square$

Based on Theorem 4.4 and Lemma 4.5, we establish the global convergence of neural TD in Algorithm 1.

**Theorem 4.6** (Convergence of Stochastic Update)**.** We set $\eta = \min\{(1-\gamma)/8, 1/\sqrt{T}\}$ in Algorithm 1. Under Assumption 4.3, the output $\widehat{Q}_{\text{out}}$ of Algorithm 1 satisfies

$$\mathbb{E}_{\text{init},\mu}\big[\big(\widehat{Q}_{\text{out}}(x) - \widehat{Q}_0(x; W^*)\big)^2\big] \leq \frac{16(B^2 + \sigma_g^2)}{(1-\gamma)^2 \sqrt{T}} + O(B^3 m^{-1/2} + B^{5/2} m^{-1/4}).$$

*Proof.* See Appendix C.6 for a detailed proof. □

As the width of the neural network $m \to \infty$, Lemma 4.2 implies that $\widehat{Q}_0(\,\cdot\,; W^*)$ is the global optimum of the MSPBE in (2.3) with a richer function class $\mathcal{F}_{B,\infty}$ to project onto. In fact, the function class $\mathcal{F}_{B,\infty} - \widehat{Q}(\,\cdot\,; W(0))$ is a subset of an RKHS with $\mathcal{H}$-norm upper bounded by $B$. Here $\widehat{Q}(\,\cdot\,; W(0))$ is defined in (3.2). See Appendix A.2 for a more detailed discussion on the representation power of $\mathcal{F}_{B,\infty}$. Therefore, if the desired Q-function $Q^\pi(\cdot)$ falls into $\mathcal{F}_{B,\infty}$, it is the global optimum of the MSPBE. In such a case, by Lemma 4.2 and Theorem 4.6, we approximately obtain $Q^\pi(\cdot) = \widehat{Q}_0(\,\cdot\,; W^*)$ through $\widehat{Q}_{\text{out}}(\cdot)$.

More generally, the following proposition quantifies the distance between $\widehat{Q}_0(\,\cdot\,; W^*)$ and $Q^\pi(\cdot)$ in the case that $Q^\pi(\cdot)$ does not fall into the function class $\mathcal{F}_{B,m}$. In particular, it states that the $\ell_2$-norm distance $\|\widehat{Q}_0(\,\cdot\,; W^*) - Q^\pi(\cdot)\|_\mu$ is upper bounded by the distance between $Q^\pi(\cdot)$ and $\mathcal{F}_{B,m}$.

**Proposition 4.7** (Convergence of Stochastic Update to $Q^\pi$)**.** It holds that $\|\widehat{Q}_0(\,\cdot\,; W^*) - Q^\pi(\cdot)\|_\mu \leq (1-\gamma)^{-1} \cdot \|\Pi_{\mathcal{F}_{B,m}} Q^\pi(\cdot) - Q^\pi(\cdot)\|_\mu$, which by Theorem 4.6 implies

$$
\mathbb{E}_{\text{init},\mu}\big[\big(\widehat{Q}_{\text{out}}(x) - Q^\pi(x)\big)^2\big] \leq \frac{32(B^2 + \sigma_g^2)}{(1-\gamma)^2\sqrt{T}} + \frac{2\mathbb{E}_{\text{init},\mu}\big[\big(\Pi_{\mathcal{F}_{B,m}} Q^\pi(x) - Q^\pi(x)\big)^2\big]}{(1-\gamma)^2}
$$
$$
+ O(B^3 m^{-1/2} + B^{5/2} m^{-1/4}).
$$

*Proof.* See Appendix B.3 for a detailed proof. □

Proposition 4.7 implies that if $Q^\pi(\cdot) \in \mathcal{F}_{B,\infty}$, then $\widehat{Q}_{\text{out}}(\cdot) \to Q^\pi(\cdot)$ as $T, m \to \infty$. In other words, neural TD converges to the global optimum of the MSPBE in (2.3), or equivalently, the MSBE in (2.2), both of which have objective value zero.

# 5 Proof Sketch

In the sequel, we sketch the proofs of Theorems 4.4 and 4.6 in Section 4.

## 5.1 Implicit Local Linearization via Overparametrization

Recall that as defined in (4.7), $\widehat{Q}_0(x; W)$ takes the form

$$
\widehat{Q}_0(x; W) = \Phi(x)^\top W,
$$
$$
\text{where } \Phi(x) = \frac{1}{\sqrt{m}} \cdot \big(\mathbb{1}\{W_1(0)^\top x > 0\}x, \ldots, \mathbb{1}\{W_m(0)^\top x > 0\}x\big) \in \mathbb{R}^{md},
$$

which is linear in the feature map $\Phi(x)$. In other words, with respect to $W$, $\widehat{Q}_0(x; W)$ linearizes the neural network $\widehat{Q}(x; W)$ defined in (3.2) locally at $W(0)$. The following lemma characterizes the difference between $\widehat{Q}(x; W(t))$, which is along the solution path of neural TD in Algorithm 1, and its local linearization $\widehat{Q}_0(x; W(t))$. In particular, we show that the error of such a local linearization diminishes to zero as $m \to \infty$. For notational simplicity, we use $\widehat{Q}_t(x)$ to denote $\widehat{Q}(x; W(t))$ in the sequel. Note that by (4.7) we have $\widehat{Q}_0(x) = \widehat{Q}(x; W(0)) = \widehat{Q}_0(x; W(0))$. Recall that $B$ is the radius of the set of feasible $W$ in (4.5).

**Lemma 5.1** (Local Linearization of Q-Function)**.** There exists a constant $c_1 > 0$ such that for any $t \in [T]$, it holds that

$$
\mathbb{E}_{\text{init},\mu}\Big[\big|\widehat{Q}_t(x) - \widehat{Q}_0\big(x; W(t)\big)\big|^2\Big] \leq 4c_1 B^3 \cdot m^{-1/2}.
$$

*Proof.* See Appendix C.1 for a detailed proof. □

As a direct consequence of Lemma 5.1, the next lemma characterizes the effect of local linearization on population semigradients. Recall that $\overline{g}(t)$ is defined in (4.10). We denote by $\overline{g}_0(t)$ the locally linearized population semigradient, which is defined by replacing $\widehat{Q}_t(x)$ in $\overline{g}(t)$ with its local linearization $\widehat{Q}_0(x; W(t))$. In other words, by (4.10), (4.7), and (4.8), we have

$$\overline{g}(t) = \mathbb{E}_\mu\big[\delta\big(x, r, x'; W(t)\big) \cdot \nabla_W \widehat{Q}\big(x; W(t)\big)\big], \tag{5.1}$$

$$\overline{g}_0(t) = \mathbb{E}_\mu\big[\delta_0\big(x, r, x'; W(t)\big) \cdot \nabla_W \widehat{Q}_0\big(x; W(t)\big)\big]. \tag{5.2}$$

**Lemma 5.2** (Local Linearization of Semigradient). Let $\overline{r}$ be the upper bound of the reward $r(x)$ for any $x \in \mathcal{X}$. There exists a constant $c_2 > 0$ such that for any $t \in [T]$, it holds that

$$\mathbb{E}_{\text{init}}\big[\|\overline{g}(t) - \overline{g}_0(t)\|_2^2\big] \leq (56c_1 B^3 + 24c_2 B + 6c_1 B\overline{r}^2) \cdot m^{-1/2}.$$

*Proof.* See Appendix C.2 for a detailed proof. $\qquad\square$

Lemmas 5.1 and 5.2 show that the error of local linearization diminishes as the degree of over-parametrization increases along $m$. As a result, we do not require the explicit local linearization in nonlinear TD [7]. Instead, we show that such an implicit local linearization suffices to ensure the global convergence of neural TD.

## 5.2 Proofs for Population Update

The characterization of the locally linearized Q-function in Lemma 5.1 and the locally linearized population semigradients in Lemma 5.2 allows us to establish the following descent lemma, which extends Lemma 3 of [6] for characterizing linear TD.

**Lemma 5.3** (Population Descent Lemma). For $\{W(t)\}_{t\in[T]}$ in Algorithm 1 with the TD update in Line 6 replaced by the population update in (4.1), it holds that

$$\|W(t+1) - W^*\|_2^2 \leq \|W(t) - W^*\|_2^2 - \big(2\eta(1-\gamma) - 8\eta^2\big) \cdot \mathbb{E}_\mu\Big[\big(\widehat{Q}_0\big(x; W(t)\big) - \widehat{Q}_0(x; W^*)\big)^2\Big]$$

$$+ \underbrace{2\eta^2 \cdot \|\overline{g}(t) - \overline{g}_0(t)\|_2^2 + 2\eta B \cdot \|\overline{g}(t) - \overline{g}_0(t)\|_2}_{\text{Error of Local Linearization}}.$$

*Proof.* See Appendix C.3 for a detailed proof. $\qquad\square$

Lemma 5.3 shows that, with a sufficiently small stepsize $\eta$, $\|W(t) - W^*\|_2$ decays at each iteration up to the error of local linearization, which is characterized by Lemma 5.2. By combining Lemmas 5.2 and 5.3 and further plugging them into a telescoping sum, we establish the convergence of $\widehat{Q}_{\text{out}}(\cdot)$ to the global optimum $\widehat{Q}_0(\cdot; W^*)$ of the MSPBE. See Appendix C.5 for a detailed proof.

## 5.3 Proofs for Stochastic Update

Recall that the stochastic semigradient $g(t)$ is defined in (4.10). In parallel with Lemma 5.3, the following lemma additionally characterizes the effect of the variance of $g(t)$, which is induced by the randomness of the current tuple $(x, r, x')$. We use the subscript $\mathbb{E}_W[\cdot]$ to denote the expectation over the randomness of the current iterate $W(t)$ conditional on the random initialization $b$ and $W(0)$. Correspondingly, $\mathbb{E}_{W,\mu}[\cdot]$ is over the randomness of both the current tuple $(x, r, x')$ and the current iterate $W(t)$ conditional on the random initialization.

**Lemma 5.4** (Stochastic Descent Lemma). For $\{W(t)\}_{t\in[T]}$ in Algorithm 1, it holds that

$$\mathbb{E}_{W,\mu}\big[\|W(t+1) - W^*\|_2^2\big]$$

$$\leq \mathbb{E}_W\big[\|W(t) - W^*\|_2^2\big] - \big(2\eta(1-\gamma) - 8\eta^2\big) \cdot \mathbb{E}_{W,\mu}\Big[\big(\widehat{Q}_0\big(x; W(t)\big) - \widehat{Q}_0(x; W^*)\big)^2\Big]$$

$$+ \underbrace{\mathbb{E}_W\big[2\eta^2 \cdot \|\overline{g}(t) - \overline{g}_0(t)\|_2^2 + 2\eta B \cdot \|\overline{g}(t) - \overline{g}_0(t)\|_2\big]}_{\text{Error of Local Linearization}} + \underbrace{\mathbb{E}_{W,\mu}\big[\eta^2 \cdot \|g(t) - \overline{g}(t)\|_2^2\big]}_{\text{Variance of Semigradient}}.$$

*Proof.* See Appendix C.4 for a detailed proof. □

To ensure the global convergence of neural TD in the presence of the variance of $g(t)$, we rescale the stepsize to be of order $T^{-1/2}$. The rest proof of Theorem 4.6 mirrors that of Theorem 4.4. See Appendix C.6 for a detailed proof.

# 6 Conclusions

In this paper we prove that neural TD converges at a sublinear rate to the global optimum of the MSPBE for policy evaluation. In particular, we show how such global convergence is enabled by the overparametrization of neural networks. Our results shed new light on the theoretical understanding of RL with neural networks, which is widely employed in practice.

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
