[Supplementary Material 1]

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

# A  Representation Power of $\mathcal{F}_{B,m}$

## A.1  Background on RKHS

We consider the following kernel function

$$K(x,y) = \int_{\mathcal{W}} \phi(x;w)\phi(y;w)p(w)dw. \qquad (A.1)$$

Here $\phi$ is a random feature map parametrized by $w$, which follows a distribution with density $p(\cdot)$ [43]. Any function in the RKHS induced by $K(\cdot,\cdot)$ takes the form

$$f_c(x) = \int_{\mathcal{W}} c(w)\phi(x;w)p(w)dw, \qquad (A.2)$$

such that each $c(\cdot)$ corresponds to a function $f_c(\cdot)$. The following lemma connects the $\mathcal{H}$-norm of $f_c(\cdot)$ to the $\ell_2$-norm of $c(\cdot)$ associated with the density $p(\cdot)$, denoted as $\|c\|_p$.

**Lemma A.1.** It holds that $\|f_c\|_{\mathcal{H}}^2 = \|c\|_p^2 = \int c(w)^2 p(w)dw$.

*Proof.* Recall if $f(x) = \int_{\mathcal{X}} a(y)K(x,y)dy$, then by the reproducing property [27], we have

$$\|f\|_{\mathcal{H}}^2 = \int_{\mathcal{X}\times\mathcal{X}} a(x)a(y)K(x,y)dxdy.$$

Now we write $f(\cdot)$ in the form of (A.2). By (A.1), we have

$$\begin{aligned}
f(x) &= \int_{\mathcal{X}} a(y)K(x,y)dy \\
&= \int_{\mathcal{X}} a(y) \int_{\mathcal{W}} \phi(x;w)\phi(y;w)p(w)dwdy \\
&= \int_{\mathcal{W}} \underbrace{\left( \int_{\mathcal{X}} a(y)\phi(y;w)dy \right)}_{c(w)} \phi(x;w)p(w)dw.
\end{aligned}$$

Thus, for $c(w) = \int_{\mathcal{X}} a(y)\phi(y;w)dy$, we have

$$\begin{aligned}
\|f\|_{\mathcal{H}}^2 &= \int_{\mathcal{X}\times\mathcal{X}} a(y)a(x)K(x,y)dxdy \\
&= \int_{\mathcal{X}\times\mathcal{X}} a(y)a(x)\left( \int_{\mathcal{W}} \phi(x;w)\phi(y;w)p(w)dw \right)dxdy \\
&= \int_{\mathcal{W}} \left( \int_{\mathcal{X}} a(y)\phi(y;w)dy \right)\left( \int_{\mathcal{X}} a(x)\phi(x;w)dx \right)p(w)dw \\
&= \int_{\mathcal{W}} c(w)^2 p(w)dw = \|c\|_p^2,
\end{aligned}$$

which completes the proof of Lemma A.1. $\qquad\square$

## A.2  $\mathcal{F}_{B,\infty}$ as RKHS

We characterize the approximate stationary point $\widehat{Q}_0(x;W^*)$ defined in Definition 4.1, which is attained by Algorithm 1 according to Theorems 4.4 and 4.6. We focus on its representation power when $m \to \infty$. We first write $\mathcal{F}_{B,m}$ in (4.5) as

$$\mathcal{F}_{B,m} = \left\{ f(x) = \widehat{Q}(x;W(0)) + \sum_{r=1}^{m} \phi_r(x)^{\top}\big(W_r - W_r(0)\big) : W \in S_B \right\}, \qquad (A.3)$$

where the feature map $\{\phi_r(x)\}_{r=1}^m$ is defined as

$$\phi_r(x) = \frac{1}{\sqrt{m}} \cdot \phi\big(x;W_r(0)\big) = \frac{1}{\sqrt{m}} \cdot \mathbb{1}\{W_r(0)^{\top}x > 0\}x \quad \text{for any } r \in [m].$$

As $m \to \infty$, the empirical distribution supported on $\{\phi_r(x)\}_{r=1}^m$, which has sample size $m$, converges to the corresponding population distribution. Therefore, from (A.3) we obtain

$$\mathcal{F}_{B,\infty} = \left\{ f(x) = f_0(x) + \int \phi(x;w)^\top \alpha(w) \cdot p(w) dw : \int \|\alpha(w)\|_2^2 \cdot p(w) dw \le B^2 \right\}.$$

Here $p(w)$ is the density of $N(0, I_d/d)$ and $f_0(x) = \lim_{m\to\infty} \widehat{Q}(x; W(0))$, which by the central limit theorem is a Gaussian process indexed by $x$. Furthermore, as discussed in Appendix A.1, $\phi(x;W)$ induces an RKHS, namely $\mathcal{H}$, which is the completion of the set of all functions that take the form

$$f(x) = \sum_{i=1}^N a_i K(x, x_i), \ \ x_i \in \mathcal{X}, \ a_i \in \mathbb{R}, \ N \in \mathbb{N},$$

$$\text{where } K(x, y) = \mathbb{E}_{w \sim N(0, I_d/d)} \big[ \mathbb{1}\{w^\top x > 0, w^\top y > 0\} x^\top y \big].$$

In particular, $\mathcal{H}$ is equipped with the inner product induced by $\langle K(\cdot, x_i), K(\cdot, x_j) \rangle_{\mathcal{H}} = K(x_i, x_j)$. [44] prove that, similar to Lemma A.1, for any $f_1(\cdot) = \int \phi(\cdot; w)^\top \alpha_1(w) \cdot p(w) dw$ and $f_2(\cdot) = \int \phi(\cdot; w)^\top \alpha_2(w) \cdot p(w) dw$, we have $f_1, f_2 \in \mathcal{H}$, and moreover, their inner product has the following equivalence

$$\langle f_1, f_2 \rangle_{\mathcal{H}} = \int \alpha_1(w)^\top \alpha_2(w) \cdot p(w) dw.$$

As a result, we have

$$\mathcal{F}_{B,\infty} = \big\{ f = f_0 + h : \|h\|_{\mathcal{H}} \le B \big\}, \tag{A.4}$$

which is known to be a rich function class [27]. As $m \to \infty$, $\widehat{Q}_0(\cdot; W^*)$ becomes the fixed-point solution to the projected Bellman equation

$$Q = \Pi_{\mathcal{F}_{B,\infty}} \mathcal{T}^\pi Q,$$

which also implies that $\widehat{Q}_0(\cdot; W^*)$ is the global optimum of the MSPBE

$$\mathbb{E}_\mu \big[ \big( Q(x) - \Pi_{\mathcal{F}_{B,\infty}} \mathcal{T}^\pi Q(x) \big)^2 \big].$$

If we further assume that the Bellman evaluation operator $\mathcal{T}^\pi$ satisfies $\mathcal{T}^\pi \widehat{Q}_0(\cdot; W^*) - f_0(\cdot) \in \mathcal{H}$ and $B$ is sufficiently large such that $\|\mathcal{T}^\pi \widehat{Q}_0(\cdot; W^*) - f_0(\cdot)\|_{\mathcal{H}} \le B$, then the projection $\Pi_{\mathcal{F}_{B,\infty}}$ reduces to identity at $\mathcal{T}^\pi \widehat{Q}_0(\cdot; W^*)$, which implies $\widehat{Q}_0(\cdot; W^*) = Q^\pi(\cdot)$ as they both solve the Bellman equation $Q = \mathcal{T}^\pi Q$. In other words, if the Bellman evaluation operator is closed with respect to $\mathcal{F}_{B,\infty}$, which up to the intercept of $f_0(\cdot)$ is a ball with radius $B$ in $\mathcal{H}$, the approximate stationary point $\widehat{Q}_0(\cdot; W^*)$ is the unique fixed-point solution to the Bellman equation or equivalently the global optimum of the MSBE

$$\mathbb{E}_\mu \big[ \big( Q(x) - \mathcal{T}^\pi Q(x) \big)^2 \big].$$

# B  Proofs for Section 4

## B.1  Proof of Lemma 4.2

*Proof.* Following the same argument for $W^\dagger$ in (4.4) and the definition of $W^*$ in (4.6), we know that $W^*$ being an approximate stationary point is equivalent to $\widehat{Q}_0(\cdot; W^*)$ being a fixed-point solution to the projected Bellman equation

$$Q = \Pi_{\mathcal{F}_{B,m}} \mathcal{T}^\pi Q. \tag{B.1}$$

Meanwhile, the Bellman evaluation operator $\mathcal{T}^\pi$ is a $\gamma$-contraction in the $\ell_2$-norm $\|\cdot\|_\mu$ with $\gamma < 1$, since

$$\mathbb{E}_{x \sim \mu} \big[ \big( \mathcal{T}^\pi Q_1(x) - \mathcal{T}^\pi Q_2(x) \big)^2 \big] = \gamma^2 \mathbb{E}_{x \sim \mu} \big[ \big( \mathbb{E}[Q_1(x') - Q_2(x') \,|\, s' \sim \mathcal{P}(\cdot \,|\, s, a), a' \sim \pi(s')] \big)^2 \big]$$

$$\le \gamma^2 \mathbb{E}_{x \sim \mu} \big[ \big( Q_1(x) - Q_2(x) \big)^2 \big],$$

where the second equality follows from Hölder's inequality and the fact that marginally $x'$ and $x$ have the same stationary distribution. Since the projection onto a convex set is nonexpansive, $\Pi_{\mathcal{F}_{B,m}} \mathcal{T}^\pi$ is also a $\gamma$-contraction. Thus, the projected Bellman equation in (B.1) has a unique fixed-point solution $\widehat{Q}_0(\cdot; W^*)$ in $\mathcal{F}_{B,m}$, which corresponds to an approximate stationary point $W^*$. $\qquad\qquad \square$

## B.2 Proof of Lemma 4.5

*Proof.* It suffices to show that $\mathbb{E}_{\text{init},\mu}[\|g(t)\|_2^2]$ is both upper bounded. By (4.10) we have

$$\mathbb{E}_{\text{init},\mu}\big[\|g(t)\|_2^2\big] = \mathbb{E}_{\text{init},\mu}\Big[\big\|\delta\big(x,r,x';W(t)\big)\cdot\nabla_W\widehat{Q}_t(x)\big\|_2^2\Big] \leq \mathbb{E}_{\text{init},\mu}\Big[\big|\delta\big(x,r,x';W(t)\big)\big|^2\Big],$$
(B.2)

where the inequality follows from the fact that, for any $W \in S_B$,

$$\|\nabla_W\widehat{Q}(x;W)\|_2 = \frac{1}{m}\sum_{r=1}^m \mathbb{1}\{W^\top x > 0\}\cdot\|x\|_2^2 \leq 1$$
(B.3)

almost everywhere. Using the fact that $x$ and $x'$ have the same marginal distribution we obtain

$$\mathbb{E}_{\text{init},\mu}\Big[\big|\delta\big(x,r,x';W(t)\big)\big|^2\Big] \leq \mathbb{E}_{\text{init},\mu}\big[3\big(\widehat{Q}_t(x)^2 + \bar{r}^2 + \widehat{Q}_t(x')^2\big)\big] = \mathbb{E}_{\text{init},\mu}[6\widehat{Q}_t(x)^2 + 3\bar{r}^2].$$
(B.4)

By (B.3), we know that $\widehat{Q}(x;W)$ is 1-Lipschitz continuous with respect to $W$. Therefore, we have

$$|\widehat{Q}_t(x) - \widehat{Q}_0(x)| \leq \|W(t) - W(0)\|_2 \leq B,$$
(B.5)

Plugging (B.5) into (B.4) and using the Cauchy-Schwarz inequality we obtain

$$\mathbb{E}_{\text{init},\mu}\Big[\big|\delta\big(x,r,x';W(t)\big)\big|^2\Big] \leq \mathbb{E}_{\text{init},\mu}[12\widehat{Q}_0(x)^2 + 12B^2 + 3\bar{r}^2].$$
(B.6)

Note that by the initialization of $\widehat{Q}_0(x)$ as defined in (3.2), we have

$$\mathbb{E}_{\text{init},\mu}[\widehat{Q}_0(x)^2] = \frac{1}{m}\sum_{r=1}^m \mathbb{E}_{\text{init}}\big[\sigma\big(W_r(0)^\top x\big)^2\big] \leq \mathbb{E}_{w\sim N(0,I_d/d)}[\|w\|_2^2] = 1.$$
(B.7)

Combining (B.2), (B.6), and (B.7) we obtain $\mathbb{E}_{\text{init},\mu}[\|g(t)\|_2^2] = O(B^2)$. Since

$$\mathbb{E}_{\text{init},\mu}\big[\|g(t) - \overline{g}(t)\|_2^2\big] = \mathbb{E}_{\text{init}}\Big[\mathbb{E}_\mu\big[\|g(t) - \overline{g}(t)\|_2^2\big]\Big] \leq \mathbb{E}_{\text{init}}\Big[\mathbb{E}_\mu\big[\|g(t)\|_2^2\big]\Big] = \mathbb{E}_{\text{init},\mu}[\|g(t)\|_2^2],$$

we conclude the proof of Lemma 4.5. □

## B.3 Proof of Proposition 4.7

*Proof.* By the triangle inequality, we have

$$\|\widehat{Q}_0(\cdot\,;W^*) - Q^\pi(\cdot)\|_\mu \leq \|\widehat{Q}_0(\cdot\,;W^*) - \Pi_{\mathcal{F}_{B,m}}Q^\pi(\cdot)\|_\mu + \|\Pi_{\mathcal{F}_{B,m}}Q^\pi(\cdot) - Q^\pi(\cdot)\|_\mu. \quad \text{(B.8)}$$

Since $Q^\pi(\cdot)$ is the fixed-point solution to the Bellman equation, we replace $Q^\pi(\cdot)$ by $\mathcal{T}^\pi Q^\pi(\cdot)$ and obtain

$$\Pi_{\mathcal{F}_{B,m}}Q^\pi(\cdot) = \Pi_{\mathcal{F}_{B,m}}\mathcal{T}^\pi Q^\pi(\cdot). \quad \text{(B.9)}$$

Meanwhile, by Lemma 4.2, $\widehat{Q}_0(\cdot\,;W^*)$ is the solution to the projected Bellman equation, that is,

$$\widehat{Q}_0(\cdot\,;W^*) = \Pi_{\mathcal{F}_{B,m}}\mathcal{T}^\pi\widehat{Q}_0(\cdot\,;W^*). \quad \text{(B.10)}$$

Combining (B.9) and (B.10), we obtain

$$\|\widehat{Q}_0(\cdot\,;W^*) - \Pi_{\mathcal{F}_{B,m}}Q^\pi(\cdot)\|_\mu = \|\Pi_{\mathcal{F}_{B,m}}\mathcal{T}^\pi\widehat{Q}_0(\cdot\,;W^*) - \Pi_{\mathcal{F}_{B,m}}\mathcal{T}^\pi Q^\pi(\cdot)\|_\mu$$
$$\leq \gamma\cdot\|\widehat{Q}_0(\cdot\,;W^*) - Q^\pi(\cdot)\|_\mu, \quad \text{(B.11)}$$

where the inequality follows from the fact that $\Pi_{\mathcal{F}_{B,m}}\mathcal{T}^\pi$ is a $\gamma$-contraction, as discussed in the proof of Lemma 4.2. Plugging (B.11) into (B.8), we obtain

$$(1-\gamma)\cdot\|\widehat{Q}_0(\cdot\,;W^*) - Q^\pi(\cdot)\|_\mu \leq \|\Pi_{\mathcal{F}_{B,m}}Q^\pi(\cdot) - Q^\pi(\cdot)\|_\mu,$$

which completes the proof of Proposition 4.7. □

## C    Proofs for Section 5

### C.1    Proof of Lemma 5.1

*Proof.* By the definition that $\widehat{Q}_t(x) = \widehat{Q}(x; W(t))$ and the definition of $\widehat{Q}_0(x; W(t))$ in (4.7), we have

$$\big|\widehat{Q}_t(x) - \widehat{Q}_0\big(x; W(t)\big)\big|$$
$$= \frac{1}{\sqrt{m}}\Big|\sum_{r=1}^{m}\big(\mathbb{1}\{W_r(t)^\top x > 0\} - \mathbb{1}\{W_r(0)^\top x > 0\}\big) \cdot b_r W_r(t)^\top x\Big|$$
$$\leq \frac{1}{\sqrt{m}}\sum_{r=1}^{m}\big|\mathbb{1}\{W_r(t)^\top x > 0\} - \mathbb{1}\{W_r(0)^\top x > 0\}\big| \cdot \big(|W_r(0)^\top x| + \|W_r(t) - W_r(0)\|_2\big),$$

$$\text{(C.1)}$$

where we use the fact that $\|x\|_2 = 1$. Note that $\mathbb{1}\{W_r(t)^\top x > 0\} \neq \mathbb{1}\{W_r(0)^\top x > 0\}$ implies

$$|W_r(0)^\top x| \leq |W_r(t)^\top x - W_r(0)^\top x| \leq \|W_r(t) - W_r(0)\|_2.$$

Thus, we obtain

$$|\mathbb{1}\{W_r(t)^\top x > 0\} - \mathbb{1}\{W_r(0)^\top x > 0\}| \leq \mathbb{1}\{|W_r(0)^\top x| \leq \|W_r(t) - W_r(0)\|_2\}. \qquad \text{(C.2)}$$

Plugging (C.2) into (C.1), we obtain the following upper bound,

$$\big|\widehat{Q}_t(x) - \widehat{Q}_0\big(x; W(t)\big)\big|$$
$$\leq \frac{1}{\sqrt{m}}\sum_{r=1}^{m}\mathbb{1}\{|W_r(0)^\top x| \leq \|W_r(t) - W_r(0)\|_2\} \cdot \big(|W_r(0)^\top x| + \|W_r(t) - W_r(0)\|_2\big)$$
$$\leq \frac{2}{\sqrt{m}}\sum_{r=1}^{m}\mathbb{1}\{|W_r(0)^\top x| \leq \|W_r(t) - W_r(0)\|_2\} \cdot \|W_r(t) - W_r(0)\|_2.$$

Here the second inequality follows from the fact that

$$\mathbb{1}\{|x| \leq y\}|x| \leq \mathbb{1}\{|x| \leq y\}y$$

for any $x$ and $y > 0$. To characterize $\mathbb{E}_{\text{init},\mu}[|\widehat{Q}_t(x) - \widehat{Q}_0(x; W(t))|^2]$, we first invoke the Cauchy-Schwarz inequality and the fact that $\|W(t) - W(0)\|_2 \leq B$, which gives

$$\big|\widehat{Q}_t(x) - \widehat{Q}_0\big(x; W(t)\big)\big|^2 \leq \frac{4B^2}{m}\sum_{r=1}^{m}\mathbb{1}\{|W_r(0)^\top x| \leq \|W_r(t) - W_r(0)\|_2\}.$$

Taking expectation on both sides, by Lemma D.1 we obtain

$$\mathbb{E}_{\text{init},\mu}\Big[\big|\widehat{Q}_t(x) - \widehat{Q}_0\big(x; W(t)\big)\big|^2\Big] \leq 4c_1 B^3 \cdot m^{-1/2}.$$

Thus, we finish the proof of Lemma 5.1.    $\square$

### C.2    Proof of Lemma 5.2

*Proof.* By the definition of $\overline{g}(t)$ and $\overline{g}_0(t)$ in (5.1) and (5.2), respectively, we have

$$\|\overline{g}(t) - \overline{g}_0(t)\|_2 = \big\|\mathbb{E}_\mu\big[\delta\big(x, r, x'; W(t)\big) \cdot \nabla_W \widehat{Q}_t(x) - \delta_0\big(x, r, x'; W(t)\big) \cdot \nabla_W \widehat{Q}_0\big(x; W(t)\big)\big]\big\|_2$$
$$\leq \Big\|\mathbb{E}_\mu\Big[\big(\delta\big(x, r, x'; W(t)\big) - \delta_0\big(x, r, x'; W(t)\big)\big) \cdot \nabla_W \widehat{Q}_t(x)$$
$$+ \delta_0\big(x, r, x'; W(t)\big) \cdot \big(\nabla_W \widehat{Q}_t(x) - \nabla_W \widehat{Q}_0\big(x; W(t)\big)\big)\Big]\Big\|_2$$
$$\leq \mathbb{E}_\mu\Big[\big|\delta\big(x, r, x'; W(t)\big) - \delta_0\big(x, r, x'; W(t)\big)\big| \qquad \text{(C.3)}$$
$$+ \big|\delta_0\big(x, r, x'; W(t)\big)\big| \cdot \big\|\nabla_W \widehat{Q}_t(x) - \nabla_W \widehat{Q}_0\big(x; W(t)\big)\big\|_2\Big].$$

Here to obtain the second inequality, we use the fact that, for any $t \in [T]$,

$$\|\nabla_W \widehat{Q}_t(x)\|_2 \leq \|x\|_2 = 1.$$

Taking expectation with respect to the random initialization on the both sides of (C.3), we obtain

$$\mathbb{E}_{\text{init}}\big[\|\overline{g}(t) - \overline{g}_0(t)\|_2^2\big]$$

$$\leq \underbrace{2\mathbb{E}_{\text{init},\mu}\Big[\big|\delta\big(x, r, x'; W(t)\big) - \delta_0\big(x, r, x'; W(t)\big)\big|^2\Big]}_{\text{(i)}} \tag{C.4}$$

$$+ 2\mathbb{E}_{\text{init}}\Big[\underbrace{\mathbb{E}_\mu\Big[\big|\delta_0\big(x, r, x'; W(t)\big)\big|^2\Big]}_{\text{(iii)}} \cdot \underbrace{\mathbb{E}_\mu\Big[\big\|\nabla_W \widehat{Q}_t(x) - \nabla_W \widehat{Q}_0\big(x; W(t)\big)\big\|_2^2\Big]}_{\text{(ii)}}\Big].$$

In the following, we characterize the three terms on the right-hand side of (C.4).

For (i) in (C.4), note that

$$\big|\delta\big(x, r, x'; W(t)\big) - \delta_0\big(x, r, x'; W(t)\big)\big|^2$$

$$= \Big|\big(\widehat{Q}_t(x) - r - \gamma \widehat{Q}_t(x')\big) - \big(\widehat{Q}_0\big(x; W(t)\big) - r - \gamma \widehat{Q}_0\big(x'; W(t)\big)\big)\Big|^2$$

$$= \Big|\big(\widehat{Q}_t(x) - \widehat{Q}_0\big(x; W(t)\big)\big) - \gamma\big(\widehat{Q}_t(x') - \widehat{Q}_0\big(x'; W(t)\big)\big)\Big|^2$$

$$\leq 2\big(\widehat{Q}_t(x) - \widehat{Q}_0\big(x; W(t)\big)\big)^2 + 2\big(\widehat{Q}_t(x') - \widehat{Q}_0\big(x'; W(t)\big)\big)^2. \tag{C.5}$$

Since $x$ and $x'$ follow the same stationary distribution $\mu$ on the right-hand side of (C.5), by Lemma 5.1 we have

$$\mathbb{E}_{\text{init},\mu}\Big[\big|\delta\big(x, r, x'; W(t)\big) - \delta_0\big(x, r, x'; W(t)\big)\big|^2\Big]$$

$$\leq 4\mathbb{E}_{\text{init},\mu}\Big[\big|\widehat{Q}_t(x) - \widehat{Q}_0\big(x; W(t)\big)\big|^2\Big] \leq 16c_1 B^3 \cdot m^{-1/2}. \tag{C.6}$$

For (ii) in (C.4), we have

$$\big\|\nabla_W \widehat{Q}_t(x) - \nabla_W \widehat{Q}_0\big(x; W(t)\big)\big\|_2^2 = \frac{1}{m}\sum_{r=1}^m (\mathbb{1}\{W_r(t)^\top > 0\} - \mathbb{1}\{W_r(0)^\top > 0\})^2 \|x\|_2^2$$

$$\leq \frac{1}{m}\sum_{r=1}^m \mathbb{1}\{|W_r(0)^\top x| \leq \|W_r(t) - W_r(0)\|_2\}, \tag{C.7}$$

where the inequality follows from (C.2) and the fact that $\|x\|_2 = 1$.

For (iii) in (C.4), we have

$$\big|\delta_0\big(x, r, x'; W(t)\big)\big|^2 \leq 3\big(\widehat{Q}_0\big(x; W(t)\big)^2 + \overline{r}^2 + \gamma^2 \widehat{Q}_0\big(x'; W(t)\big)^2\big). \tag{C.8}$$

To obtain an upper bound of the right-hand side of (C.8), we use the fact that

$$\big|\widehat{Q}_0\big(x; W(t)\big) - \widehat{Q}_0(x)\big| \leq \|W(t) - W(0)\|_2 \cdot \|x\|_2 \leq B,$$

which follows from (4.7), and obtain

$$\mathbb{E}_\mu\big[\widehat{Q}_0\big(x; W(t)\big)^2\big] = \mathbb{E}_\mu\Big[\big(\widehat{Q}_0(x) + \widehat{Q}_0\big(x; W(t)\big) - \widehat{Q}_0(x)\big)^2\Big] \leq 2\mathbb{E}_\mu[\widehat{Q}_0(x)^2] + 2B^2.$$

Since $x$ and $x'$ follow the same stationary distribution $\mu$ on the right-hand side of (C.8) and $|\gamma| < 1$, we have

$$\mathbb{E}_\mu\Big[\big|\delta_0\big(x, r, x'; W(t)\big)\big|^2\Big] \leq 12\mathbb{E}_\mu[\widehat{Q}_0(x)^2] + 12B^2 + 3\overline{r}^2. \tag{C.9}$$

Plugging (C.6), (C.7), and (C.9) into (C.4), we obtain

$$\mathbb{E}_{\text{init}}\big[\|\overline{g}(t) - \overline{g}_0(t)\|_2^2\big] \le 32c_1 B^3 \cdot m^{-1/2}$$

$$+ 2\mathbb{E}_{\text{init}}\Big[\big(12\mathbb{E}_\mu[\widehat{Q}_0(x)^2] + 12B^2 + 3\overline{r}^2\big) \cdot \Big(\frac{1}{m}\sum_{r=1}^m \mathbb{1}\{|W_r(0)^\top x| \le \|W_r(t) - W_r(0)\|_2\}\Big)\Big].$$

Invoking Lemmas D.1 and D.2, we obtain

$$\mathbb{E}_{\text{init}}\big[\|\overline{g}(t) - \overline{g}_0(t)\|_2^2\big] \le (56c_1 B^3 + 24c_2 B + 6c_1 B\overline{r}^2) \cdot m^{-1/2},$$

which finishes the proof of Lemma 5.2. $\qquad\qquad\square$

## C.3  Proof of Lemma 5.3

*Proof.* Recall that

$$\overline{g}(t) = \mathbb{E}_\mu\big[\delta\big(x, r, x'; W(t)\big) \cdot \nabla_W \widehat{Q}\big(x; W(t)\big)\big], \tag{C.10}$$

$$\overline{g}_0(t) = \mathbb{E}_\mu\big[\delta_0\big(x, r, x'; W(t)\big) \cdot \nabla_W \widehat{Q}_0\big(x; W(t)\big)\big]. \tag{C.11}$$

We denote the locally linearized population semigradient $\overline{g}_0(t)$ evaluated at the approximate stationary point $W^*$ by

$$\overline{g}_0^* = \mathbb{E}_\mu[\delta_0(x, r, x'; W^*) \cdot \nabla_W \widehat{Q}_0(x; W^*)]. \tag{C.12}$$

For any $W(t)$ ($t \in [T]$), by the convexity of $S_B$, we have

$$\|W(t+1) - W^*\|_2^2 = \big\|\Pi_{S_B}\big(W(t) - \eta \cdot \overline{g}(t)\big) - \Pi_{S_B}(W^* - \eta \cdot \overline{g}_0^*)\big\|_2^2$$

$$\le \big\|\big(W(t) - \eta \cdot \overline{g}(t)\big) - (W^* - \eta \cdot \overline{g}_0^*)\big\|_2^2$$

$$= \|W(t) - W^*\|_2^2 - 2\eta \cdot \big(\overline{g}(t) - \overline{g}_0^*\big)^\top \big(W(t) - W^*\big) + \eta^2 \cdot \|\overline{g}(t) - \overline{g}_0^*\|_2^2. \tag{C.13}$$

We decompose the inner product $(\overline{g}(t) - \overline{g}_0^*)^\top (W(t) - W^*)$ on the right-hand side of (C.13) into two terms,

$$\big(\overline{g}(t) - \overline{g}_0^*\big)^\top \big(W(t) - W^*\big) = \big(\overline{g}_0(t) - \overline{g}_0^*\big)^\top \big(W(t) - W^*\big) + \big(\overline{g}(t) - \overline{g}_0(t)\big)^\top \big(W(t) - W^*\big)$$

$$\ge \big(\overline{g}_0(t) - \overline{g}_0^*\big)^\top \big(W(t) - W^*\big) - B \cdot \|\overline{g}(t) - \overline{g}_0(t)\|_2. \tag{C.14}$$

It remains to characterize the first term $(\overline{g}_0(t) - \overline{g}_0^*)^\top (W(t) - W^*)$ on the right-hand side of (C.14), since the second term $\|\overline{g}(t) - \overline{g}_0(t)\|_2$ is characterized by Lemma 5.2. Note that by (C.11) and (C.12), we have

$$\overline{g}_0(t) - \overline{g}_0^* = \mathbb{E}_\mu\Big[\big(\delta_0\big(x, r, x'; W(t)\big) - \delta_0(x, r, x'; W^*)\big) \cdot \nabla_W \widehat{Q}_0\big(x; W(0)\big)\Big], \tag{C.15}$$

where we use the following consequence of (4.7),

$$\nabla_W \widehat{Q}_0\big(x; W(0)\big) = \nabla_W \widehat{Q}_0(x; W^*).$$

Moreover, by (4.8) it holds that

$$\delta_0\big(x, r, x'; W(t)\big) - \delta_0(x, r, x'; W^*)$$

$$= \big(\widehat{Q}_0\big(x; W(t)\big) - \widehat{Q}_0(x; W^*)\big) - \gamma\big(\widehat{Q}_0\big(x'; W(t)\big) - \widehat{Q}_0(x'; W^*)\big). \tag{C.16}$$

Combining (4.7), (C.15), and (C.16), we have

$$\big(\overline{g}_0(t) - \overline{g}_0^*\big)^\top \big(W(t) - W^*\big)$$

$$= \mathbb{E}_\mu\Big[\big(\delta_0\big(x, r, x'; W(t)\big) - \delta_0(x, r, x'; W^*)\big) \cdot \big(\nabla_W \widehat{Q}_0\big(x; W(0)\big)^\top \big(W(t) - W^*\big)\big)\Big]$$

$$= \mathbb{E}_\mu\Big[\big(\widehat{Q}_0\big(x; W(t)\big) - \widehat{Q}_0(x; W^*)\big)^2$$

$$\qquad - \gamma\big(\widehat{Q}_0\big(x; W(t)\big) - \widehat{Q}_0(x; W^*)\big) \cdot \big(\widehat{Q}_0\big(x'; W(t)\big) - \widehat{Q}_0(x'; W^*)\big)\Big]$$

$$\ge (1 - \gamma) \cdot \mathbb{E}_\mu\Big[\big(\widehat{Q}_0\big(x; W(t)\big) - \widehat{Q}_0(x; W^*)\big)^2\Big], \tag{C.17}$$

where the last inequality is from the fact that $x$ and $x'$ have the same marginal distribution under $\mu$ and therefore by the Cauchy-Schwarz inequality,

$$
\begin{aligned}
\mathbb{E}_\mu & \left[ \left( \widehat{Q}_0(x; W(t)) - \widehat{Q}_0(x; W^*) \right) \cdot \left( \widehat{Q}_0(x'; W(t)) - \widehat{Q}_0(x'; W^*) \right) \right] \\
& \leq \mathbb{E}_\mu \left[ \left( \widehat{Q}_0(x; W(t)) - \widehat{Q}_0(x; W^*) \right)^2 \right]^{1/2} \cdot \mathbb{E}_\mu \left[ \left( \widehat{Q}_0(x'; W(t)) - \widehat{Q}_0(x'; W^*) \right)^2 \right]^{1/2} \\
& = \mathbb{E}_\mu \left[ \left( \widehat{Q}_0(x; W(t)) - \widehat{Q}_0(x; W^*) \right)^2 \right].
\end{aligned}
$$

Inequality (C.17) is the key to our convergence result. It shows that the locally linearized population semigradient update $\overline{g}_0(t)$ is one-point monotonic to the approximate stationary point $W^*$.

Also, for $\|\overline{g}(t) - \overline{g}_0^*\|_2^2$ on the right-hand side of (C.13), we have

$$
\|\overline{g}(t) - \overline{g}_0^*\|_2^2 \leq 2\|\overline{g}_0(t) - \overline{g}_0^*\|_2^2 + 2\|\overline{g}(t) - \overline{g}_0(t)\|_2^2. \tag{C.18}
$$

For the first term on the right-hand side of (C.18), by (C.15), (C.16), and the Cauchy-Schwarz inequality, we have

$$
\begin{aligned}
\|\overline{g}_0(t) - \overline{g}_0^*\|_2^2 & = \left\| \mathbb{E}_\mu \left[ \left( \delta_0(x, r, x'; W(t)) - \delta_0(x, r, x'; W^*) \right) \cdot \nabla_W \widehat{Q}_0(x; W(0)) \right] \right\|^2 \\
& \leq \mathbb{E}_\mu \left[ \left( \widehat{Q}_0(x; W(t)) - \widehat{Q}_0(x; W^*) - \gamma \widehat{Q}_0(x'; W(t)) + \gamma \widehat{Q}_0(x'; W^*) \right)^2 \right] \\
& \leq 4 \mathbb{E}_\mu \left[ \left( \widehat{Q}_0(x; W(t)) - \widehat{Q}_0(x; W^*) \right)^2 \right], \tag{C.19}
\end{aligned}
$$

where the first inequality follows from the fact

$$
\left\| \nabla_W \widehat{Q}_0(x; W(0)) \right\|_2 \leq \|x\|_2 = 1.
$$

Plugging (C.17), (C.18), and (C.19) into (C.13), we finish the proof of Lemma 5.3. $\qquad\square$

## C.4 Proof of Lemma 5.4

*Proof.* For any $W(t)$ ($t \in [T]$), by the convexity of $S_B$, (4.10), and (C.12), we have

$$
\begin{aligned}
\|W(t+1) - W^*\|_2^2 & = \left\| \Pi_{S_B} \left( W(t) - \eta \cdot g(t) \right) - \Pi_{S_B}(W^* - \eta \cdot \overline{g}_0^*) \right\|_2^2 \\
& \leq \left\| \left( W(t) - \eta \cdot g(t) \right) - (W^* - \eta \cdot \overline{g}_0^*) \right\|_2^2 \\
& = \|W(t) - W^*\|_2^2 - 2\eta \cdot \left( g(t) - \overline{g}_0^* \right)^\top \left( W(t) - W^* \right) + \eta^2 \cdot \|g(t) - \overline{g}_0^*\|_2^2.
\end{aligned}
$$

Taking expectation on both sides conditional on $W(t)$, we obtain

$$
\begin{aligned}
\mathbb{E}_\mu & \left[ \|W(t+1) - W^*\|_2^2 \,\big|\, W(t) \right] \\
& \leq \|W(t) - W^*\|_2^2 - 2\eta \cdot \left( \overline{g}(t) - \overline{g}_0^* \right)^\top \left( W(t) - W^* \right) + \eta^2 \cdot \mathbb{E}_\mu \left[ \|g(t) - \overline{g}_0^*\|_2^2 \,\big|\, W(t) \right]. \tag{C.20}
\end{aligned}
$$

For the inner product $\left( \overline{g}(t) - \overline{g}_0^* \right)^\top \left( W(t) - W^* \right)$ on the right-hand side of (C.20), it follows from (C.14) and (C.17) that

$$
\left( \overline{g}(t) - \overline{g}_0^* \right)^\top \left( W(t) - W^* \right) \geq (1 - \gamma) \cdot \mathbb{E}_\mu \left[ \left( \widehat{Q}_0(x; W(t)) - \widehat{Q}_0(x; W^*) \right)^2 \right] - B \cdot \|\overline{g}(t) - \overline{g}_0(t)\|_2.
$$

Meanwhile, for $\mathbb{E}_\mu[\|g(t) - \overline{g}_0^*\|_2^2 \,|\, W(t)]$ on the right-hand side of (C.20), we have the decomposition

$$
\begin{aligned}
\mathbb{E}_\mu & \left[ \|g(t) - \overline{g}_0^*\|_2^2 \,\big|\, W(t) \right] = \|\overline{g}(t) - \overline{g}_0^*\|_2^2 + \mathbb{E}_\mu \left[ \|g(t) - \overline{g}(t)\|_2^2 \,\big|\, W(t) \right] \\
& \leq 8 \mathbb{E}_\mu \left[ \left( \widehat{Q}_0(x; W(t)) - \widehat{Q}_0(x; W^*) \right)^2 \,\big|\, W(t) \right] + 2\|\overline{g}(t) - \overline{g}_0(t)\|_2^2 + \mathbb{E}_\mu \left[ \|g(t) - \overline{g}(t)\|_2^2 \,\big|\, W(t) \right],
\end{aligned}
$$

where the inequality follows from (C.18) and (C.19). Taking expectation on the both sides of (C.20) with respect to $W(t)$, we complete the proof of Lemma 5.4. $\qquad\square$

## C.5  Proof of Theorem 4.4

*Proof.* By Lemma 5.2 we have

$$\mathbb{E}_{\text{init}}\big[\|\overline{g}(t) - \overline{g}_0(t)\|_2^2\big] = O(B^3 m^{-1/2}), \tag{C.21}$$

$$\mathbb{E}_{\text{init}}\big[B \cdot \|\overline{g}(t) - \overline{g}_0(t)\|_2\big] = O(B^{5/2} m^{-1/4}). \tag{C.22}$$

Setting $\eta = (1-\gamma)/8$ in Algorithm 1, by (C.21), (C.22), and Lemma 5.3, we have

$$\mathbb{E}_{\text{init},\mu}\Big[\big(\widehat{Q}_0(x; W(t)) - \widehat{Q}_0(x; W^*)\big)^2\Big] = \frac{\mathbb{E}_{\text{init}}\big[\|W(t) - W^*\|_2^2 - \|W(t+1) - W^*\|_2^2\big]}{(1-\gamma)^2/8} \tag{C.23}$$
$$+ O(B^3 m^{-1/2} + B^{5/2} m^{-1/4}).$$

Telescoping (C.23) for $t = 0, \ldots, T-1$, we obtain

$$\frac{1}{T}\sum_{t=0}^{T-1}\mathbb{E}_{\text{init},\mu}\Big[\big(\widehat{Q}_0(x; W(t)) - \widehat{Q}_0(x; W^*)\big)^2\Big]$$

$$= \frac{\mathbb{E}_{\text{init}}\big[\|W(0) - W^*\|^2 - \|W(T) - W^*\|^2\big]}{T(1-\gamma)^2/8} + O(B^3 m^{-1/2} + B^{5/2} m^{-1/4})$$

$$\le \frac{8B^2}{T(1-\gamma)^2} + O(B^3 m^{-1/2} + B^{5/2} m^{-1/4}).$$

Recall that as define in (4.7), $\widehat{Q}_0(\cdot\,; W)$ is linear in $W$. By Jensen's inequality, we have

$$\mathbb{E}_{\text{init},\mu}\big[\big(\widehat{Q}_0(x; \overline{W}) - \widehat{Q}_0(x; W^*)\big)^2\big] \le \frac{8B^2}{T(1-\gamma)^2} + O(B^3 m^{-1/2} + B^{5/2} m^{-1/4}). \tag{C.24}$$

Next we characterize the output $\widehat{Q}_{\text{out}}(\cdot) = \widehat{Q}(\cdot\,; \overline{W})$ of Algorithm 1. Since $S_B$ is convex and $\overline{W} \in S_B$, by Lemma 5.1 we have

$$\mathbb{E}_{\text{init},\mu}\big[\big(\widehat{Q}_0(x; \overline{W}) - \widehat{Q}_0(x; W^*)\big)^2\big] = O(B^3 m^{-1/2}). \tag{C.25}$$

Using the Cauchy-Schwarz inequality we have

$$\mathbb{E}_{\text{init},\mu}\big[\big(\widehat{Q}_{\text{out}}(x) - \widehat{Q}_0(x; W^*)\big)^2\big]$$

$$\le \mathbb{E}_{\text{init},\mu}\big[2\big(\widehat{Q}(x; \overline{W}) - \widehat{Q}_0(x; \overline{W})\big)^2 + 2\big(\widehat{Q}_0(x; \overline{W}) - \widehat{Q}_0(x; W^*)\big)^2\big],$$

into which we plugging (C.24) and (C.25) and obtain

$$\mathbb{E}_{\text{init},\mu}\big[\big(\widehat{Q}_{\text{out}}(x) - \widehat{Q}_0(x; W^*)\big)^2\big] \le \frac{16B^2}{T(1-\gamma)^2} + O(B^3 m^{-1/2} + B^{5/2} m^{-1/4}), \tag{C.26}$$

which completes the proof of Theorem 4.4. □

## C.6  Proof of Theorem 4.6

*Proof.* Similar to (C.23), by Lemmas 4.5, 5.2, and 5.4 we have

$$\mathbb{E}_{\text{init},\mu}\Big[\big(\widehat{Q}_0(x; W(t)) - \widehat{Q}_0(x; W^*)\big)^2\Big]$$

$$\le \frac{\mathbb{E}_{\text{init}}\big[\|W(t) - W^*\|_2^2\big] - \mathbb{E}_{\text{init}}\big[\|W(t+1) - W^*\|_2^2\big] + \eta^2 \cdot \sigma_g^2}{2\eta(1-\gamma) - 8\eta^2}$$

$$+ O(B^3 m^{-1/2} + B^{5/2} m^{-1/4}). \tag{C.27}$$

Telescoping (C.27) for $t = 0, \ldots, T-1$, by $\eta^2 \le 1/T$ we have

$$\frac{1}{T}\sum_{t=0}^{T-1}\mathbb{E}_{\text{init},\mu}\Big[\big(\widehat{Q}_0(x; W(t)) - \widehat{Q}_0(x; W^*)\big)^2\Big]$$

$$\le \frac{\mathbb{E}_{\text{init}}\big[\|W(t) - W^*\|_2^2\big] + \sigma_g^2}{T \cdot \big(2\eta(1-\gamma) - 8\eta^2\big)} + O(B^3 m^{-1/2} + B^{5/2} m^{-1/4})$$

$$\le \frac{B^2 + \sigma_g^2}{\sqrt{T}} \cdot \frac{1}{\sqrt{T} \cdot \big(2\eta(1-\gamma) - 8\eta^2\big)} + O(B^3 m^{-1/2} + B^{5/2} m^{-1/4}), \tag{C.28}$$

where $\eta = \min\{1/\sqrt{T}, (1-\gamma)/8\}$. Note that when $T \geq (8/(1-\gamma))^2$, we have $\eta = 1/\sqrt{T}$ and

$$\sqrt{T} \cdot \left(2\eta(1-\gamma) - 8\eta^2\right) = 2(1-\gamma) - 8/\sqrt{T} \geq 1 - \gamma.$$

Meanwhile, when $T < (8/(1-\gamma))^2$, we have $\eta = (1-\gamma)/8$ and

$$\sqrt{T} \cdot \left(2\eta(1-\gamma) - 8\eta^2\right) = \sqrt{T} \cdot (1-\gamma)^2/8 \geq (1-\gamma)^2/8.$$

Since $|1-\gamma| < 1$, we obtain that for any $T \in \mathbb{N}$,

$$\frac{1}{\sqrt{T} \cdot \left(2\eta(1-\gamma) - 8\eta^2\right)} \leq \frac{8}{(1-\gamma)^2}. \tag{C.29}$$

Similar to (C.24) and (C.26), by combining (C.28) and (C.29) with Lemma 5.1, we obtain

$$\mathbb{E}_{\mathrm{init},\mu}\left[\left(\widehat{Q}_{\mathrm{out}}(x) - \widehat{Q}_0(x; W^*)\right)^2\right] \leq \frac{16(B^2 + \sigma_g^2)}{\sqrt{T} \cdot (1-\gamma)^2} + O(B^3 m^{-1/2} + B^{5/2} m^{-1/4}),$$

which completes the proof of Theorem 4.6. $\qquad\square$

## D  Auxiliary Lemmas

Under Assumption 4.3, we establish the following auxiliary lemmas on the random initialization $W(0)$ and the stationary distribution $\mu$, which plays a key role in quantifying the error of local linearization.

**Lemma D.1.** There exists a constant $c_1 > 0$ such that for any random vector $W$ with $\|W - W(0)\|_2 \leq B$, it holds that

$$\mathbb{E}_{\mathrm{init},\mu}\left[\frac{1}{m}\sum_{r=1}^m \mathbb{1}\{|W_r(0)^\top x| \leq \|W_r - W_r(0)\|_2\}\right] \leq c_1 B \cdot m^{-1/2}. \tag{D.1}$$

*Proof.* By Assumption 4.3, we have

$$\mathbb{E}_{\mathrm{init},\mu}\left[\frac{1}{m}\sum_{r=1}^m \mathbb{1}\{|W_r(0)^\top x| \leq \|W_r - W_r(0)\|_2\}\right]$$

$$\leq \mathbb{E}_{\mathrm{init}}\left[\frac{1}{m}\sum_{r=1}^m c_0 \cdot \|W_r - W_r(0)\|_2/\|W_r(0)\|_2\right]. \tag{D.2}$$

Applying Hölder's inequality to the right-hand side, we obtain

$$\mathbb{E}_{\mathrm{init},\mu}\left[\frac{1}{m}\sum_{r=1}^m \mathbb{1}\{|W_r(0)^\top x| \leq \|W_r - W_r(0)\|_2\}\right]$$

$$\leq c_0/m \cdot \mathbb{E}_{\mathrm{init}}\left[\left(\sum_{r=1}^m \|W_r - W_r(0)\|_2^2\right)^{1/2} \cdot \left(\sum_{r=1}^m \frac{1}{\|W_r(0)\|_2^2}\right)^{1/2}\right]$$

$$\leq c_0 B \cdot m^{-1/2} \cdot \mathbb{E}_{w \sim N(0, I_d/d)}\left[1/\|w\|_2^2\right]^{1/2}, \tag{D.3}$$

where the second inequality follows from

$$\mathbb{E}_{\mathrm{init}}\left[\left(\sum_{r=1}^m \frac{1}{\|W_r(0)\|_2^2}\right)^{1/2}\right] \leq \mathbb{E}_{\mathrm{init}}\left[\sum_{r=1}^m \frac{1}{\|W_r(0)\|_2^2}\right]^{1/2} = \sqrt{m} \cdot \mathbb{E}_{w \sim N(0, I_d/d)}\left[1/\|w\|_2^2\right]^{1/2}. \tag{D.4}$$

Setting $c_1 = c_0 \cdot \mathbb{E}_{w \sim N(0, I_d/d)}[1/\|w\|_2^2]^{1/2}$, we complete the proof of Lemma D.1. $\qquad\square$

**Lemma D.2.** There exists a constant $c_2 > 0$ such that for any random vector $W$ with $\|W - W(0)\|_2 \leq B$, it holds that

$$\mathbb{E}_{\mathrm{init}}\left[\mathbb{E}_\mu\left[\widehat{Q}_0(x)^2\right] \cdot \mathbb{E}_\mu\left[\frac{1}{m}\sum_{r=1}^m \mathbb{1}\{|W_r(0)^\top x| \leq \|W_r - W_r(0)\|_2\}\right]\right] \leq c_2 B \cdot m^{-1/2}. \tag{D.5}$$

*Proof.* By the definition of $\widehat{Q}_0(x) = \widehat{Q}_0(x; W(0))$ in (4.7), we have

$$\mathbb{E}_\mu\big[\widehat{Q}_0(x)^2\big] = 1/m \cdot \mathbb{E}_\mu\Big[\sum_{r=1}^m \sigma\big(W_r(0)^\top x\big)^2 + \sum_{r\neq s} b_r b_s \sigma\big(W_r(0)^\top x\big)\sigma\big(W_s(0)^\top x\big)\Big].$$

Following the same derivation of (D.2) and (D.3), we have

$$\mathbb{E}_{\text{init}}\Big[\mathbb{E}_\mu\big[\widehat{Q}_0(x)^2\big] \cdot \mathbb{E}_\mu\Big[\frac{1}{m}\sum_{r=1}^m \mathbb{1}\{|W_r(0)^\top x| \leq \|W_r - W_r(0)\|_2\}\Big]\Big]$$

$$\leq \mathbb{E}_{\text{init}}\Big[1/m \cdot \mathbb{E}_\mu\Big[\sum_{r=1}^m \sigma\big(W_r(0)^\top x\big)^2 + \sum_{r\neq s} b_r b_s \sigma\big(W_r(0)^\top x\big)\sigma\big(W_s(0)^\top x\big)\Big]$$

$$\cdot c_0/m \cdot \Big(\sum_{r=1}^m \|W_r - W_r(0)\|_2^2\Big)^{1/2} \cdot \Big(\sum_{r=1}^m \frac{1}{\|W_r(0)\|_2^2}\Big)^{1/2}\Big].$$

Note that $b_r$ and $b_s$ are independent of $W(0)$ and $\mathbb{E}_{\text{init}}[b_r b_s] = 0$. Thus, we obtain

$$\mathbb{E}_{\text{init}}\Big[\mathbb{E}_\mu\big[\widehat{Q}_0(x)^2\big] \cdot \mathbb{E}_\mu\Big[\frac{1}{m}\sum_{r=1}^m \mathbb{1}\{|W_r(0)^\top x| \leq \|W_r - W_r(0)\|_2\}\Big]\Big]$$

$$\leq c_0 B/m^2 \cdot \mathbb{E}_{\text{init}}\Big[\mathbb{E}_\mu\Big[\sum_{r=1}^m \sigma\big(W_r(0)^\top x\big)^2\Big] \cdot \Big(\sum_{r=1}^m \frac{1}{\|W_r(0)\|_2^2}\Big)^{1/2}\Big].$$

By the definition of $\sigma(W_r(0)^\top x)$ and the fact that $\|x\|_2 = 1$, we have

$$\mathbb{E}_\mu\Big[\sum_{r=1}^m \sigma\big(W_r(0)^\top x\big)^2\Big] \leq \sum_{r=1}^m \|W_r(0)\|_2^2.$$

Hence, it holds that

$$\mathbb{E}_{\text{init}}\Big[\mathbb{E}_\mu\big[\widehat{Q}_0(x)^2\big] \cdot \mathbb{E}_\mu\Big[\frac{1}{m}\sum_{r=1}^m \mathbb{1}\{|W_r(0)^\top x| \leq \|W_r - W_r(0)\|_2\}\Big]\Big]$$

$$\leq c_0 B/m^2 \cdot \mathbb{E}_{\text{init}}\Big[\Big(\sum_{r=1}^m \|W_r(0)\|_2^2\Big) \cdot \Big(\sum_{r=1}^m \frac{1}{\|W_r(0)\|_2^2}\Big)^{1/2}\Big]$$

$$\leq c_0 B/m^2 \cdot \mathbb{E}_{\text{init}}\Big[\Big(\sum_{r=1}^m \|W_r(0)\|_2^2\Big)^2\Big]^{1/2} \cdot \mathbb{E}_{\text{init}}\Big[\sum_{r=1}^m \frac{1}{\|W_r(0)\|_2^2}\Big]^{1/2}. \qquad (\text{D}.6)$$

By (D.4) and the fact that

$$\mathbb{E}_{\text{init}}\Big[\Big(\sum_{r=1}^m \|W_r(0)\|_2^2\Big)^2\Big] = m \cdot \mathbb{E}_{w\sim N(0, I_d/d)}\big[\|w\|_2^4\big] + m(m-1) \cdot \mathbb{E}_{w\sim N(0, I_d/d)}\big[\|w\|_2^2\big]^2 = O(m^2),$$

the right-hand side of (D.6) is $O(Bm^{-1/2})$. Setting

$$c_2 = c_0 \cdot \Big(\mathbb{E}_{w\sim N(0, I_d/d)}\big[\|w\|_2^4\big] + \mathbb{E}_{w\sim N(0, I_d/d)}\big[\|w\|_2^2\big]^2\Big)^{1/2} \cdot \mathbb{E}_{w\sim N(0, I_d/d)}\big[1/\|w\|_2^2\big]^{1/2},$$

we complete the proof of Lemma D.2. $\qquad\qquad\square$

[Supplementary Material 2]

# A Representation Power of $\mathcal{F}_{B,m}$

## A.1 Background on RKHS

We consider the following kernel function

$$K(x, y) = \int_{\mathcal{W}} \phi(x; w)\phi(y; w)p(w)dw. \tag{A.1}$$

Here $\phi$ is a random feature map parametrized by $w$, which follows a distribution with density $p(\cdot)$ [43]. Any function in the RKHS induced by $K(\cdot, \cdot)$ takes the form

$$f_c(x) = \int_{\mathcal{W}} c(w)\phi(x; w)p(w)dw, \tag{A.2}$$

such that each $c(\cdot)$ corresponds to a function $f_c(\cdot)$. The following lemma connects the $\mathcal{H}$-norm of $f_c(\cdot)$ to the $\ell_2$-norm of $c(\cdot)$ associated with the density $p(\cdot)$, denoted as $\|c\|_p$.

**Lemma A.1.** It holds that $\|f_c\|_{\mathcal{H}}^2 = \|c\|_p^2 = \int c(w)^2 p(w)dw$.

*Proof.* Recall if $f(x) = \int_{\mathcal{X}} a(y)K(x, y)dy$, then by the reproducing property [27], we have

$$\|f\|_{\mathcal{H}}^2 = \int_{\mathcal{X} \times \mathcal{X}} a(x)a(y)K(x, y)dxdy.$$

Now we write $f(\cdot)$ in the form of (A.2). By (A.1), we have

$$\begin{aligned}
f(x) &= \int_{\mathcal{X}} a(y)K(x, y)dy \\
&= \int_{\mathcal{X}} a(y) \int_{\mathcal{W}} \phi(x; w)\phi(y; w)p(w)dwdy \\
&= \int_{\mathcal{W}} \underbrace{\left( \int_{\mathcal{X}} a(y)\phi(y; w)dy \right)}_{c(w)} \phi(x; w)p(w)dw.
\end{aligned}$$

Thus, for $c(w) = \int_{\mathcal{X}} a(y)\phi(y; w)dy$, we have

$$\begin{aligned}
\|f\|_{\mathcal{H}}^2 &= \int_{\mathcal{X} \times \mathcal{X}} a(y)a(x)K(x, y)dxdy \\
&= \int_{\mathcal{X} \times \mathcal{X}} a(y)a(x) \left( \int_{\mathcal{W}} \phi(x; w)\phi(y; w)p(w)dw \right)dxdy \\
&= \int_{\mathcal{W}} \left( \int_{\mathcal{X}} a(y)\phi(y; w)dy \right)\left( \int_{\mathcal{X}} a(x)\phi(x; w)dx \right)p(w)dw \\
&= \int_{\mathcal{W}} c(w)^2 p(w)dw = \|c\|_p^2,
\end{aligned}$$

which completes the proof of Lemma A.1. $\qquad\square$

## A.2 $\mathcal{F}_{B,\infty}$ as RKHS

We characterize the approximate stationary point $\widehat{Q}_0(x; W^*)$ defined in Definition 4.1, which is attained by Algorithm 1 according to Theorems 4.4 and 4.6. We focus on its representation power when $m \to \infty$. We first write $\mathcal{F}_{B,m}$ in (4.5) as

$$\mathcal{F}_{B,m} = \left\{ f(x) = \widehat{Q}(x; W(0)) + \sum_{r=1}^{m} \phi_r(x)^{\top}\left( W_r - W_r(0) \right) : W \in S_B \right\}, \tag{A.3}$$

where the feature map $\{\phi_r(x)\}_{r=1}^{m}$ is defined as

$$\phi_r(x) = \frac{1}{\sqrt{m}} \cdot \phi\left( x; W_r(0) \right) = \frac{1}{\sqrt{m}} \cdot \mathbb{1}\{W_r(0)^{\top}x > 0\}x \text{ for any } r \in [m].$$

As $m \to \infty$, the empirical distribution supported on $\{\phi_r(x)\}_{r=1}^m$, which has sample size $m$, converges to the corresponding population distribution. Therefore, from (A.3) we obtain

$$\mathcal{F}_{B,\infty} = \left\{ f(x) = f_0(x) + \int \phi(x;w)^\top \alpha(w) \cdot p(w)dw : \int \|\alpha(w)\|_2^2 \cdot p(w)dw \le B^2 \right\}.$$

Here $p(w)$ is the density of $N(0, I_d/d)$ and $f_0(x) = \lim_{m \to \infty} \widehat{Q}(x; W(0))$, which by the central limit theorem is a Gaussian process indexed by $x$. Furthermore, as discussed in Appendix A.1, $\phi(x; W)$ induces an RKHS, namely $\mathcal{H}$, which is the completion of the set of all functions that take the form

$$f(x) = \sum_{i=1}^N a_i K(x, x_i), \ \ x_i \in \mathcal{X}, \ a_i \in \mathbb{R}, \ N \in \mathbb{N},$$

$$\text{where } K(x, y) = \mathbb{E}_{w \sim N(0, I_d/d)}\big[\mathbb{1}\{w^\top x > 0, w^\top y > 0\}x^\top y\big].$$

In particular, $\mathcal{H}$ is equipped with the inner product induced by $\langle K(\cdot, x_i), K(\cdot, x_j)\rangle_{\mathcal{H}} = K(x_i, x_j)$. [44] prove that, similar to Lemma A.1, for any $f_1(\cdot) = \int \phi(\cdot; w)^\top \alpha_1(w) \cdot p(w)dw$ and $f_2(\cdot) = \int \phi(\cdot; w)^\top \alpha_2(w) \cdot p(w)dw$, we have $f_1, f_2 \in \mathcal{H}$, and moreover, their inner product has the following equivalence

$$\langle f_1, f_2\rangle_{\mathcal{H}} = \int \alpha_1(w)^\top \alpha_2(w) \cdot p(w)dw.$$

As a result, we have

$$\mathcal{F}_{B,\infty} = \big\{ f = f_0 + h : \|h\|_{\mathcal{H}} \le B \big\}, \tag{A.4}$$

which is known to be a rich function class [27]. As $m \to \infty$, $\widehat{Q}_0(\cdot; W^*)$ becomes the fixed-point solution to the projected Bellman equation

$$Q = \Pi_{\mathcal{F}_{B,\infty}} \mathcal{T}^\pi Q,$$

which also implies that $\widehat{Q}_0(\cdot; W^*)$ is the global optimum of the MSPBE

$$\mathbb{E}_\mu\big[ (Q(x) - \Pi_{\mathcal{F}_{B,\infty}} \mathcal{T}^\pi Q(x))^2 \big].$$

If we further assume that the Bellman evaluation operator $\mathcal{T}^\pi$ satisfies $\mathcal{T}^\pi \widehat{Q}_0(\cdot; W^*) - f_0(\cdot) \in \mathcal{H}$ and $B$ is sufficiently large such that $\|\mathcal{T}^\pi \widehat{Q}_0(\cdot; W^*) - f_0(\cdot)\|_{\mathcal{H}} \le B$, then the projection $\Pi_{\mathcal{F}_{B,\infty}}$ reduces to identity at $\mathcal{T}^\pi \widehat{Q}_0(\cdot; W^*)$, which implies $\widehat{Q}_0(\cdot; W^*) = Q^\pi(\cdot)$ as they both solve the Bellman equation $Q = \mathcal{T}^\pi Q$. In other words, if the Bellman evaluation operator is closed with respect to $\mathcal{F}_{B,\infty}$, which up to the intercept of $f_0(\cdot)$ is a ball with radius $B$ in $\mathcal{H}$, the approximate stationary point $\widehat{Q}_0(\cdot; W^*)$ is the unique fixed-point solution to the Bellman equation or equivalently the global optimum of the MSBE

$$\mathbb{E}_\mu\big[ (Q(x) - \mathcal{T}^\pi Q(x))^2 \big].$$

# B  Proofs for Section 4

## B.1  Proof of Lemma 4.2

*Proof.* Following the same argument for $W^\dagger$ in (4.4) and the definition of $W^*$ in (4.6), we know that $W^*$ being an approximate stationary point is equivalent to $\widehat{Q}_0(\cdot; W^*)$ being a fixed-point solution to the projected Bellman equation

$$Q = \Pi_{\mathcal{F}_{B,m}} \mathcal{T}^\pi Q. \tag{B.1}$$

Meanwhile, the Bellman evaluation operator $\mathcal{T}^\pi$ is a $\gamma$-contraction in the $\ell_2$-norm $\|\cdot\|_\mu$ with $\gamma < 1$, since

$$\mathbb{E}_{x \sim \mu}\big[(\mathcal{T}^\pi Q_1(x) - \mathcal{T}^\pi Q_2(x))^2\big] = \gamma^2 \mathbb{E}_{x \sim \mu}\big[\big(\mathbb{E}[Q_1(x') - Q_2(x') \,|\, s' \sim \mathcal{P}(\cdot \,|\, s, a), a' \sim \pi(s')]\big)^2\big]$$
$$\le \gamma^2 \mathbb{E}_{x \sim \mu}\big[(Q_1(x) - Q_2(x))^2\big],$$

where the second equality follows from Hölder's inequality and the fact that marginally $x'$ and $x$ have the same stationary distribution. Since the projection onto a convex set is nonexpansive, $\Pi_{\mathcal{F}_{B,m}} \mathcal{T}^\pi$ is also a $\gamma$-contraction. Thus, the projected Bellman equation in (B.1) has a unique fixed-point solution $\widehat{Q}_0(\cdot; W^*)$ in $\mathcal{F}_{B,m}$, which corresponds to an approximate stationary point $W^*$. $\qquad\square$

## B.2 Proof of Lemma 4.5

*Proof.* It suffices to show that $\mathbb{E}_{\text{init},\mu}[\|g(t)\|_2^2]$ is both upper bounded. By (4.10) we have

$$\mathbb{E}_{\text{init},\mu}\big[\|g(t)\|_2^2\big] = \mathbb{E}_{\text{init},\mu}\Big[\big\|\delta\big(x,r,x';W(t)\big)\cdot\nabla_W\widehat{Q}_t(x)\big\|_2^2\Big] \leq \mathbb{E}_{\text{init},\mu}\Big[\big|\delta\big(x,r,x';W(t)\big)\big|^2\Big],$$
(B.2)

where the inequality follows from the fact that, for any $W \in S_B$,

$$\|\nabla_W\widehat{Q}(x;W)\|_2 = \frac{1}{m}\sum_{r=1}^m \mathbb{1}\{W^\top x > 0\}\cdot\|x\|_2^2 \leq 1$$
(B.3)

almost everywhere. Using the fact that $x$ and $x'$ have the same marginal distribution we obtain

$$\mathbb{E}_{\text{init},\mu}\big[\big|\delta\big(x,r,x';W(t)\big)\big|^2\big] \leq \mathbb{E}_{\text{init},\mu}\big[3\big(\widehat{Q}_t(x)^2 + \bar{r}^2 + \widehat{Q}_t(x')^2\big)\big] = \mathbb{E}_{\text{init},\mu}[6\widehat{Q}_t(x)^2 + 3\bar{r}^2].$$
(B.4)

By (B.3), we know that $\widehat{Q}(x;W)$ is 1-Lipschitz continuous with respect to $W$. Therefore, we have

$$|\widehat{Q}_t(x) - \widehat{Q}_0(x)| \leq \|W(t) - W(0)\|_2 \leq B,$$
(B.5)

Plugging (B.5) into (B.4) and using the Cauchy-Schwarz inequality we obtain

$$\mathbb{E}_{\text{init},\mu}\big[\big|\delta\big(x,r,x';W(t)\big)\big|^2\big] \leq \mathbb{E}_{\text{init},\mu}[12\widehat{Q}_0(x)^2 + 12B^2 + 3\bar{r}^2].$$
(B.6)

Note that by the initialization of $\widehat{Q}_0(x)$ as defined in (3.2), we have

$$\mathbb{E}_{\text{init},\mu}[\widehat{Q}_0(x)^2] = \frac{1}{m}\sum_{r=1}^m \mathbb{E}_{\text{init}}\big[\sigma\big(W_r(0)^\top x\big)^2\big] \leq \mathbb{E}_{w\sim N(0,I_d/d)}[\|w\|_2^2] = 1.$$
(B.7)

Combining (B.2), (B.6), and (B.7) we obtain $\mathbb{E}_{\text{init},\mu}\big[\|g(t)\|_2^2\big] = O(B^2)$. Since

$$\mathbb{E}_{\text{init},\mu}\big[\|g(t) - \overline{g}(t)\|_2^2\big] = \mathbb{E}_{\text{init}}\Big[\mathbb{E}_\mu\big[\|g(t) - \overline{g}(t)\|_2^2\big]\Big] \leq \mathbb{E}_{\text{init}}\Big[\mathbb{E}_\mu\big[\|g(t)\|_2^2\big]\Big] = \mathbb{E}_{\text{init},\mu}[\|g(t)\|_2^2],$$

we conclude the proof of Lemma 4.5. $\qquad\square$

## B.3 Proof of Proposition 4.7

*Proof.* By the triangle inequality, we have

$$\|\widehat{Q}_0(\cdot;W^*) - Q^\pi(\cdot)\|_\mu \leq \|\widehat{Q}_0(\cdot;W^*) - \Pi_{\mathcal{F}_{B,m}}Q^\pi(\cdot)\|_\mu + \|\Pi_{\mathcal{F}_{B,m}}Q^\pi(\cdot) - Q^\pi(\cdot)\|_\mu. \quad \text{(B.8)}$$

Since $Q^\pi(\cdot)$ is the fixed-point solution to the Bellman equation, we replace $Q^\pi(\cdot)$ by $\mathcal{T}^\pi Q^\pi(\cdot)$ and obtain

$$\Pi_{\mathcal{F}_{B,m}}Q^\pi(\cdot) = \Pi_{\mathcal{F}_{B,m}}\mathcal{T}^\pi Q^\pi(\cdot).$$
(B.9)

Meanwhile, by Lemma 4.2, $\widehat{Q}_0(\cdot;W^*)$ is the solution to the projected Bellman equation, that is,

$$\widehat{Q}_0(\cdot;W^*) = \Pi_{\mathcal{F}_{B,m}}\mathcal{T}^\pi\widehat{Q}_0(\cdot;W^*).$$
(B.10)

Combining (B.9) and (B.10), we obtain

$$\|\widehat{Q}_0(\cdot;W^*) - \Pi_{\mathcal{F}_{B,m}}Q^\pi(\cdot)\|_\mu = \|\Pi_{\mathcal{F}_{B,m}}\mathcal{T}^\pi\widehat{Q}_0(\cdot;W^*) - \Pi_{\mathcal{F}_{B,m}}\mathcal{T}^\pi Q^\pi(\cdot)\|_\mu$$
$$\leq \gamma\cdot\|\widehat{Q}_0(\cdot;W^*) - Q^\pi(\cdot)\|_\mu, \quad \text{(B.11)}$$

where the inequality follows from the fact that $\Pi_{\mathcal{F}_{B,m}}\mathcal{T}^\pi$ is a $\gamma$-contraction, as discussed in the proof of Lemma 4.2. Plugging (B.11) into (B.8), we obtain

$$(1-\gamma)\cdot\|\widehat{Q}_0(\cdot;W^*) - Q^\pi(\cdot)\|_\mu \leq \|\Pi_{\mathcal{F}_{B,m}}Q^\pi(\cdot) - Q^\pi(\cdot)\|_\mu,$$

which completes the proof of Proposition 4.7. $\qquad\square$

## C  Proofs for Section 5

### C.1  Proof of Lemma 5.1

*Proof.* By the definition that $\widehat{Q}_t(x) = \widehat{Q}(x; W(t))$ and the definition of $\widehat{Q}_0(x; W(t))$ in (4.7), we have

$$
\begin{aligned}
\big|\widehat{Q}_t(x) &- \widehat{Q}_0\big(x; W(t)\big)\big| \\
&= \frac{1}{\sqrt{m}}\Big|\sum_{r=1}^{m}\big(\mathbb{1}\{W_r(t)^\top x > 0\} - \mathbb{1}\{W_r(0)^\top x > 0\}\big)\cdot b_r W_r(t)^\top x\Big| \\
&\leq \frac{1}{\sqrt{m}}\sum_{r=1}^{m}|\mathbb{1}\{W_r(t)^\top x > 0\} - \mathbb{1}\{W_r(0)^\top x > 0\}|\cdot \big(|W_r(0)^\top x| + \|W_r(t) - W_r(0)\|_2\big),
\end{aligned}
$$
$$\tag{C.1}$$

where we use the fact that $\|x\|_2 = 1$. Note that $\mathbb{1}\{W_r(t)^\top x > 0\} \neq \mathbb{1}\{W_r(0)^\top x > 0\}$ implies

$$
|W_r(0)^\top x| \leq |W_r(t)^\top x - W_r(0)^\top x| \leq \|W_r(t) - W_r(0)\|_2.
$$

Thus, we obtain

$$
|\mathbb{1}\{W_r(t)^\top x > 0\} - \mathbb{1}\{W_r(0)^\top x > 0\}| \leq \mathbb{1}\{|W_r(0)^\top x| \leq \|W_r(t) - W_r(0)\|_2\}. \tag{C.2}
$$

Plugging (C.2) into (C.1), we obtain the following upper bound,

$$
\begin{aligned}
\big|\widehat{Q}_t(x) &- \widehat{Q}_0\big(x; W(t)\big)\big| \\
&\leq \frac{1}{\sqrt{m}}\sum_{r=1}^{m}\mathbb{1}\{|W_r(0)^\top x| \leq \|W_r(t) - W_r(0)\|_2\}\cdot\big(|W_r(0)^\top x| + \|W_r(t) - W_r(0)\|_2\big) \\
&\leq \frac{2}{\sqrt{m}}\sum_{r=1}^{m}\mathbb{1}\{|W_r(0)^\top x| \leq \|W_r(t) - W_r(0)\|_2\}\cdot\|W_r(t) - W_r(0)\|_2.
\end{aligned}
$$

Here the second inequality follows from the fact that

$$
\mathbb{1}\{|x| \leq y\}|x| \leq \mathbb{1}\{|x| \leq y\}y
$$

for any $x$ and $y > 0$. To characterize $\mathbb{E}_{\text{init},\mu}[|\widehat{Q}_t(x) - \widehat{Q}_0(x; W(t))|^2]$, we first invoke the Cauchy-Schwarz inequality and the fact that $\|W(t) - W(0)\|_2 \leq B$, which gives

$$
\big|\widehat{Q}_t(x) - \widehat{Q}_0\big(x; W(t)\big)\big|^2 \leq \frac{4B^2}{m}\sum_{r=1}^{m}\mathbb{1}\{|W_r(0)^\top x| \leq \|W_r(t) - W_r(0)\|_2\}.
$$

Taking expectation on both sides, by Lemma D.1 we obtain

$$
\mathbb{E}_{\text{init},\mu}\Big[\big|\widehat{Q}_t(x) - \widehat{Q}_0\big(x; W(t)\big)\big|^2\Big] \leq 4c_1 B^3 \cdot m^{-1/2}.
$$

Thus, we finish the proof of Lemma 5.1. $\qquad\square$

### C.2  Proof of Lemma 5.2

*Proof.* By the definition of $\overline{g}(t)$ and $\overline{g}_0(t)$ in (5.1) and (5.2), respectively, we have

$$
\begin{aligned}
\|\overline{g}(t) - \overline{g}_0(t)\|_2 &= \big\|\mathbb{E}_\mu\big[\delta\big(x, r, x'; W(t)\big)\cdot\nabla_W\widehat{Q}_t(x) - \delta_0\big(x, r, x'; W(t)\big)\cdot\nabla_W\widehat{Q}_0\big(x; W(t)\big)\big]\big\|_2 \\
&\leq \Big\|\mathbb{E}_\mu\Big[\big(\delta\big(x, r, x'; W(t)\big) - \delta_0\big(x, r, x'; W(t)\big)\big)\cdot\nabla_W\widehat{Q}_t(x) \\
&\qquad + \delta_0\big(x, r, x'; W(t)\big)\cdot\big(\nabla_W\widehat{Q}_t(x) - \nabla_W\widehat{Q}_0\big(x; W(t)\big)\big)\Big]\Big\|_2 \\
&\leq \mathbb{E}_\mu\Big[\big|\delta\big(x, r, x'; W(t)\big) - \delta_0\big(x, r, x'; W(t)\big)\big| \\
&\qquad + \big|\delta_0\big(x, r, x'; W(t)\big)\big|\cdot\big\|\nabla_W\widehat{Q}_t(x) - \nabla_W\widehat{Q}_0\big(x; W(t)\big)\big\|_2\Big].
\end{aligned}
$$
$$\tag{C.3}$$

Here to obtain the second inequality, we use the fact that, for any $t \in [T]$,

$$\|\nabla_W \widehat{Q}_t(x)\|_2 \leq \|x\|_2 = 1.$$

Taking expectation with respect to the random initialization on the both sides of (C.3), we obtain

$$
\begin{aligned}
&\mathbb{E}_{\text{init}}\big[\|\overline{g}(t) - \overline{g}_0(t)\|_2^2\big] \\
&\leq \underbrace{2\mathbb{E}_{\text{init},\mu}\Big[\big|\delta\big(x,r,x';W(t)\big) - \delta_0\big(x,r,x';W(t)\big)\big|^2\Big]}_{(i)} \qquad\qquad\qquad\text{(C.4)} \\
&\quad + 2\mathbb{E}_{\text{init}}\Big[\underbrace{\mathbb{E}_\mu\Big[\big|\delta_0\big(x,r,x';W(t)\big)\big|^2\Big]}_{(iii)} \cdot \underbrace{\mathbb{E}_\mu\Big[\big\|\nabla_W \widehat{Q}_t(x) - \nabla_W \widehat{Q}_0\big(x;W(t)\big)\big\|_2^2\Big]}_{(ii)}\Big].
\end{aligned}
$$

In the following, we characterize the three terms on the right-hand side of (C.4).

For (i) in (C.4), note that

$$
\begin{aligned}
&\big|\delta\big(x,r,x';W(t)\big) - \delta_0\big(x,r,x';W(t)\big)\big|^2 \\
&= \Big|\big(\widehat{Q}_t(x) - r - \gamma\widehat{Q}_t(x')\big) - \big(\widehat{Q}_0\big(x;W(t)\big) - r - \gamma\widehat{Q}_0\big(x';W(t)\big)\big)\Big|^2 \\
&= \Big|\big(\widehat{Q}_t(x) - \widehat{Q}_0\big(x;W(t)\big)\big) - \gamma\big(\widehat{Q}_t(x') - \widehat{Q}_0\big(x';W(t)\big)\big)\Big|^2 \\
&\leq 2\big(\widehat{Q}_t(x) - \widehat{Q}_0\big(x;W(t)\big)\big)^2 + 2\big(\widehat{Q}_t(x') - \widehat{Q}_0\big(x';W(t)\big)\big)^2. \qquad\text{(C.5)}
\end{aligned}
$$

Since $x$ and $x'$ follow the same stationary distribution $\mu$ on the right-hand side of (C.5), by Lemma 5.1 we have

$$
\begin{aligned}
&\mathbb{E}_{\text{init},\mu}\Big[\big|\delta\big(x,r,x';W(t)\big) - \delta_0\big(x,r,x';W(t)\big)\big|^2\Big] \\
&\qquad\leq 4\mathbb{E}_{\text{init},\mu}\Big[\big|\widehat{Q}_t(x) - \widehat{Q}_0\big(x;W(t)\big)\big|^2\Big] \leq 16c_1 B^3 \cdot m^{-1/2}. \qquad\text{(C.6)}
\end{aligned}
$$

For (ii) in (C.4), we have

$$
\begin{aligned}
\big\|\nabla_W \widehat{Q}_t(x) - \nabla_W \widehat{Q}_0\big(x;W(t)\big)\big\|_2^2 &= \frac{1}{m}\sum_{r=1}^m (\mathbb{1}\{W_r(t)^\top > 0\} - \mathbb{1}\{W_r(0)^\top > 0\})^2 \|x\|_2^2 \\
&\leq \frac{1}{m}\sum_{r=1}^m \mathbb{1}\{|W_r(0)^\top x| \leq \|W_r(t) - W_r(0)\|_2\}, \qquad\text{(C.7)}
\end{aligned}
$$

where the inequality follows from (C.2) and the fact that $\|x\|_2 = 1$.

For (iii) in (C.4), we have

$$\big|\delta_0\big(x,r,x';W(t)\big)\big|^2 \leq 3\Big(\widehat{Q}_0\big(x;W(t)\big)^2 + \overline{r}^2 + \gamma^2 \widehat{Q}_0\big(x';W(t)\big)^2\Big). \qquad\text{(C.8)}$$

To obtain an upper bound of the right-hand side of (C.8), we use the fact that

$$\big|\widehat{Q}_0\big(x;W(t)\big) - \widehat{Q}_0(x)\big| \leq \|W(t) - W(0)\|_2 \cdot \|x\|_2 \leq B,$$

which follows from (4.7), and obtain

$$\mathbb{E}_\mu\big[\widehat{Q}_0\big(x;W(t)\big)^2\big] = \mathbb{E}_\mu\Big[\big(\widehat{Q}_0(x) + \widehat{Q}_0\big(x;W(t)\big) - \widehat{Q}_0(x)\big)^2\Big] \leq 2\mathbb{E}_\mu[\widehat{Q}_0(x)^2] + 2B^2.$$

Since $x$ and $x'$ follow the same stationary distribution $\mu$ on the right-hand side of (C.8) and $|\gamma| < 1$, we have

$$\mathbb{E}_\mu\Big[\big|\delta_0\big(x,r,x';W(t)\big)\big|^2\Big] \leq 12\mathbb{E}_\mu[\widehat{Q}_0(x)^2] + 12B^2 + 3\overline{r}^2. \qquad\text{(C.9)}$$

Plugging (C.6), (C.7), and (C.9) into (C.4), we obtain

$$\mathbb{E}_{\text{init}}\big[\|\bar{g}(t) - \bar{g}_0(t)\|_2^2\big] \le 32c_1B^3 \cdot m^{-1/2}$$

$$+ 2\mathbb{E}_{\text{init}}\Big[\big(12\mathbb{E}_\mu[\widehat{Q}_0(x)^2] + 12B^2 + 3\bar{r}^2\big) \cdot \Big(\frac{1}{m}\sum_{r=1}^m \mathbb{1}\{|W_r(0)^\top x| \le \|W_r(t) - W_r(0)\|_2\}\Big)\Big].$$

Invoking Lemmas D.1 and D.2, we obtain

$$\mathbb{E}_{\text{init}}\big[\|\bar{g}(t) - \bar{g}_0(t)\|_2^2\big] \le (56c_1B^3 + 24c_2B + 6c_1B\bar{r}^2) \cdot m^{-1/2},$$

which finishes the proof of Lemma 5.2. $\qquad\square$

## C.3 Proof of Lemma 5.3

*Proof.* Recall that

$$\bar{g}(t) = \mathbb{E}_\mu\big[\delta\big(x, r, x'; W(t)\big) \cdot \nabla_W \widehat{Q}\big(x; W(t)\big)\big], \tag{C.10}$$

$$\bar{g}_0(t) = \mathbb{E}_\mu\big[\delta_0\big(x, r, x'; W(t)\big) \cdot \nabla_W \widehat{Q}_0\big(x; W(t)\big)\big]. \tag{C.11}$$

We denote the locally linearized population semigradient $\bar{g}_0(t)$ evaluated at the approximate stationary point $W^*$ by

$$\bar{g}_0^* = \mathbb{E}_\mu[\delta_0(x, r, x'; W^*) \cdot \nabla_W \widehat{Q}_0(x; W^*)]. \tag{C.12}$$

For any $W(t)$ ($t \in [T]$), by the convexity of $S_B$, we have

$$\|W(t+1) - W^*\|_2^2 = \big\|\Pi_{S_B}\big(W(t) - \eta \cdot \bar{g}(t)\big) - \Pi_{S_B}(W^* - \eta \cdot \bar{g}_0^*)\big\|_2^2$$

$$\le \big\|\big(W(t) - \eta \cdot \bar{g}(t)\big) - (W^* - \eta \cdot \bar{g}_0^*)\big\|_2^2$$

$$= \|W(t) - W^*\|_2^2 - 2\eta \cdot \big(\bar{g}(t) - \bar{g}_0^*\big)^\top \big(W(t) - W^*\big) + \eta^2 \cdot \|\bar{g}(t) - \bar{g}_0^*\|_2^2. \tag{C.13}$$

We decompose the inner product $(\bar{g}(t) - \bar{g}_0^*)^\top (W(t) - W^*)$ on the right-hand side of (C.13) into two terms,

$$\big(\bar{g}(t) - \bar{g}_0^*\big)^\top \big(W(t) - W^*\big) = \big(\bar{g}_0(t) - \bar{g}_0^*\big)^\top \big(W(t) - W^*\big) + \big(\bar{g}(t) - \bar{g}_0(t)\big)^\top \big(W(t) - W^*\big)$$

$$\ge \big(\bar{g}_0(t) - \bar{g}_0^*\big)^\top \big(W(t) - W^*\big) - B \cdot \|\bar{g}(t) - \bar{g}_0(t)\|_2. \tag{C.14}$$

It remains to characterize the first term $(\bar{g}_0(t) - \bar{g}_0^*)^\top (W(t) - W^*)$ on the right-hand side of (C.14), since the second term $\|\bar{g}(t) - \bar{g}_0(t)\|_2$ is characterized by Lemma 5.2. Note that by (C.11) and (C.12), we have

$$\bar{g}_0(t) - \bar{g}_0^* = \mathbb{E}_\mu\Big[\big(\delta_0\big(x, r, x'; W(t)\big) - \delta_0(x, r, x'; W^*)\big) \cdot \nabla_W \widehat{Q}_0\big(x; W(0)\big)\Big], \tag{C.15}$$

where we use the following consequence of (4.7),

$$\nabla_W \widehat{Q}_0\big(x; W(0)\big) = \nabla_W \widehat{Q}_0(x; W^*).$$

Moreover, by (4.8) it holds that

$$\delta_0\big(x, r, x'; W(t)\big) - \delta_0(x, r, x'; W^*)$$

$$= \Big(\widehat{Q}_0\big(x; W(t)\big) - \widehat{Q}_0(x; W^*)\Big) - \gamma\Big(\widehat{Q}_0\big(x'; W(t)\big) - \widehat{Q}_0(x'; W^*)\Big). \tag{C.16}$$

Combining (4.7), (C.15), and (C.16), we have

$$\big(\bar{g}_0(t) - \bar{g}_0^*\big)^\top \big(W(t) - W^*\big)$$

$$= \mathbb{E}_\mu\Big[\big(\delta_0\big(x, r, x'; W(t)\big) - \delta_0(x, r, x'; W^*)\big) \cdot \Big(\nabla_W \widehat{Q}_0\big(x; W(0)\big)^\top \big(W(t) - W^*\big)\Big)\Big]$$

$$= \mathbb{E}_\mu\Big[\Big(\widehat{Q}_0\big(x; W(t)\big) - \widehat{Q}_0(x; W^*)\Big)^2$$

$$- \gamma\Big(\widehat{Q}_0\big(x; W(t)\big) - \widehat{Q}_0(x; W^*)\Big) \cdot \Big(\widehat{Q}_0\big(x'; W(t)\big) - \widehat{Q}_0(x'; W^*)\Big)\Big]$$

$$\ge (1 - \gamma) \cdot \mathbb{E}_\mu\Big[\Big(\widehat{Q}_0\big(x; W(t)\big) - \widehat{Q}_0(x; W^*)\Big)^2\Big], \tag{C.17}$$

where the last inequality is from the fact that $x$ and $x'$ have the same marginal distribution under $\mu$ and therefore by the Cauchy-Schwarz inequality,

$$
\mathbb{E}_\mu\left[\left(\widehat{Q}_0\big(x;W(t)\big)-\widehat{Q}_0(x;W^*)\right)\cdot\left(\widehat{Q}_0\big(x';W(t)\big)-\widehat{Q}_0(x';W^*)\right)\right]
$$

$$
\leq \mathbb{E}_\mu\left[\left(\widehat{Q}_0\big(x;W(t)\big)-\widehat{Q}_0(x;W^*)\right)^2\right]^{1/2}\cdot\mathbb{E}_\mu\left[\left(\widehat{Q}_0\big(x';W(t)\big)-\widehat{Q}_0(x';W^*)\right)^2\right]^{1/2}
$$

$$
= \mathbb{E}_\mu\left[\left(\widehat{Q}_0\big(x;W(t)\big)-\widehat{Q}_0(x;W^*)\right)^2\right].
$$

Inequality (C.17) is the key to our convergence result. It shows that the locally linearized population semigradient update $\overline{g}_0(t)$ is one-point monotonic to the approximate stationary point $W^*$.

Also, for $\|\overline{g}(t)-\overline{g}_0^*\|_2^2$ on the right-hand side of (C.13), we have

$$
\|\overline{g}(t)-\overline{g}_0^*\|_2^2 \leq 2\|\overline{g}_0(t)-\overline{g}_0^*\|_2^2 + 2\|\overline{g}(t)-\overline{g}_0(t)\|_2^2. \tag{C.18}
$$

For the first term on the right-hand side of (C.18), by (C.15), (C.16), and the Cauchy-Schwarz inequality, we have

$$
\|\overline{g}_0(t)-\overline{g}_0^*\|_2^2 = \left\|\mathbb{E}_\mu\left[\left(\delta_0\big(x,r,x';W(t)\big)-\delta_0(x,r,x';W^*)\right)\cdot\nabla_W\widehat{Q}_0\big(x;W(0)\big)\right]\right\|^2
$$

$$
\leq \mathbb{E}_\mu\left[\left(\widehat{Q}_0\big(x;W(t)\big)-\widehat{Q}_0(x;W^*)-\gamma\widehat{Q}_0\big(x';W(t)\big)+\gamma\widehat{Q}_0(x';W^*)\right)^2\right]
$$

$$
\leq 4\mathbb{E}_\mu\left[\left(\widehat{Q}_0\big(x;W(t)\big)-\widehat{Q}_0(x;W^*)\right)^2\right], \tag{C.19}
$$

where the first inequality follows from the fact

$$
\left\|\nabla_W\widehat{Q}_0\big(x;W(0)\big)\right\|_2 \leq \|x\|_2 = 1.
$$

Plugging (C.17), (C.18), and (C.19) into (C.13), we finish the proof of Lemma 5.3. $\qquad\square$

## C.4  Proof of Lemma 5.4

*Proof.* For any $W(t)$ ($t\in[T]$), by the convexity of $S_B$, (4.10), and (C.12), we have

$$
\|W(t+1)-W^*\|_2^2 = \left\|\Pi_{S_B}\big(W(t)-\eta\cdot g(t)\big)-\Pi_{S_B}(W^*-\eta\cdot\overline{g}_0^*)\right\|_2^2
$$

$$
\leq \left\|\big(W(t)-\eta\cdot g(t)\big)-(W^*-\eta\cdot\overline{g}_0^*)\right\|_2^2
$$

$$
= \|W(t)-W^*\|_2^2 - 2\eta\cdot\big(g(t)-\overline{g}_0^*\big)^\top\big(W(t)-W^*\big) + \eta^2\cdot\|g(t)-\overline{g}_0^*\|_2^2.
$$

Taking expectation on both sides conditional on $W(t)$, we obtain

$$
\mathbb{E}_\mu\big[\|W(t+1)-W^*\|_2^2\,\big|\,W(t)\big]
$$

$$
\leq \|W(t)-W^*\|_2^2 - 2\eta\cdot\big(\overline{g}(t)-\overline{g}_0^*\big)^\top\big(W(t)-W^*\big) + \eta^2\cdot\mathbb{E}_\mu\big[\|g(t)-\overline{g}_0^*\|_2^2\,\big|\,W(t)\big]. \tag{C.20}
$$

For the inner product $\big(\overline{g}(t)-\overline{g}_0^*\big)^\top\big(W(t)-W^*\big)$ on the right-hand side of (C.20), it follows from (C.14) and (C.17) that

$$
\big(\overline{g}(t)-\overline{g}_0^*\big)^\top\big(W(t)-W^*\big) \geq (1-\gamma)\cdot\mathbb{E}_\mu\left[\left(\widehat{Q}_0\big(x;W(t)\big)-\widehat{Q}_0(x;W^*)\right)^2\right] - B\cdot\|\overline{g}(t)-\overline{g}_0(t)\|_2.
$$

Meanwhile, for $\mathbb{E}_\mu[\|g(t)-\overline{g}_0^*\|_2^2\,|\,W(t)]$ on the right-hand side of (C.20), we have the decomposition

$$
\mathbb{E}_\mu\big[\|g(t)-\overline{g}_0^*\|_2^2\,\big|\,W(t)\big] = \|\overline{g}(t)-\overline{g}_0^*\|_2^2 + \mathbb{E}_\mu\big[\|g(t)-\overline{g}(t)\|_2^2\,\big|\,W(t)\big]
$$

$$
\leq 8\mathbb{E}_\mu\left[\left(\widehat{Q}_0\big(x;W(t)\big)-\widehat{Q}_0(x;W^*)\right)^2\,\Big|\,W(t)\right] + 2\|\overline{g}(t)-\overline{g}_0(t)\|_2^2 + \mathbb{E}_\mu\big[\|g(t)-\overline{g}(t)\|_2^2\,\big|\,W(t)\big],
$$

where the inequality follows from (C.18) and (C.19). Taking expectation on the both sides of (C.20) with respect to $W(t)$, we complete the proof of Lemma 5.4. $\qquad\square$

## C.5 Proof of Theorem 4.4

*Proof.* By Lemma 5.2 we have

$$\mathbb{E}_{\text{init}}\big[\|\overline{g}(t) - \overline{g}_0(t)\|_2^2\big] = O(B^3 m^{-1/2}), \tag{C.21}$$

$$\mathbb{E}_{\text{init}}\big[B \cdot \|\overline{g}(t) - \overline{g}_0(t)\|_2\big] = O(B^{5/2} m^{-1/4}). \tag{C.22}$$

Setting $\eta = (1-\gamma)/8$ in Algorithm 1, by (C.21), (C.22), and Lemma 5.3, we have

$$\mathbb{E}_{\text{init},\mu}\Big[\big(\widehat{Q}_0(x; W(t)) - \widehat{Q}_0(x; W^*)\big)^2\Big] = \frac{\mathbb{E}_{\text{init}}\big[\|W(t) - W^*\|_2^2 - \|W(t+1) - W^*\|_2^2\big]}{(1-\gamma)^2/8} \tag{C.23}$$
$$+ O(B^3 m^{-1/2} + B^{5/2} m^{-1/4}).$$

Telescoping (C.23) for $t = 0, \ldots, T-1$, we obtain

$$\frac{1}{T}\sum_{t=0}^{T-1}\mathbb{E}_{\text{init},\mu}\Big[\big(\widehat{Q}_0(x; W(t)) - \widehat{Q}_0(x; W^*)\big)^2\Big]$$
$$= \frac{\mathbb{E}_{\text{init}}\big[\|W(0) - W^*\|^2 - \|W(T) - W^*\|^2\big]}{T(1-\gamma)^2/8} + O(B^3 m^{-1/2} + B^{5/2} m^{-1/4})$$
$$\leq \frac{8B^2}{T(1-\gamma)^2} + O(B^3 m^{-1/2} + B^{5/2} m^{-1/4}).$$

Recall that as define in (4.7), $\widehat{Q}_0(\cdot\,; W)$ is linear in $W$. By Jensen's inequality, we have

$$\mathbb{E}_{\text{init},\mu}\big[\big(\widehat{Q}_0(x; \overline{W}) - \widehat{Q}_0(x; W^*)\big)^2\big] \leq \frac{8B^2}{T(1-\gamma)^2} + O(B^3 m^{-1/2} + B^{5/2} m^{-1/4}). \tag{C.24}$$

Next we characterize the output $\widehat{Q}_{\text{out}}(\cdot) = \widehat{Q}(\cdot\,; \overline{W})$ of Algorithm 1. Since $S_B$ is convex and $\overline{W} \in S_B$, by Lemma 5.1 we have

$$\mathbb{E}_{\text{init},\mu}\big[\big(\widehat{Q}_0(x; \overline{W}) - \widehat{Q}_0(x; W^*)\big)^2\big] = O(B^3 m^{-1/2}). \tag{C.25}$$

Using the Cauchy-Schwarz inequality we have

$$\mathbb{E}_{\text{init},\mu}\big[\big(\widehat{Q}_{\text{out}}(x) - \widehat{Q}_0(x; W^*)\big)^2\big]$$
$$\leq \mathbb{E}_{\text{init},\mu}\big[2\big(\widehat{Q}(x; \overline{W}) - \widehat{Q}_0(x; \overline{W})\big)^2 + 2\big(\widehat{Q}_0(x; \overline{W}) - \widehat{Q}_0(x; W^*)\big)^2\big],$$

into which we plugging (C.24) and (C.25) and obtain

$$\mathbb{E}_{\text{init},\mu}\big[\big(\widehat{Q}_{\text{out}}(x) - \widehat{Q}_0(x; W^*)\big)^2\big] \leq \frac{16B^2}{T(1-\gamma)^2} + O(B^3 m^{-1/2} + B^{5/2} m^{-1/4}), \tag{C.26}$$

which completes the proof of Theorem 4.4. □

## C.6 Proof of Theorem 4.6

*Proof.* Similar to (C.23), by Lemmas 4.5, 5.2, and 5.4 we have

$$\mathbb{E}_{\text{init},\mu}\Big[\big(\widehat{Q}_0(x; W(t)) - \widehat{Q}_0(x; W^*)\big)^2\Big]$$
$$\leq \frac{\mathbb{E}_{\text{init}}\big[\|W(t) - W^*\|_2^2\big] - \mathbb{E}_{\text{init}}\big[\|W(t+1) - W^*\|_2^2\big] + \eta^2 \cdot \sigma_g^2}{2\eta(1-\gamma) - 8\eta^2}$$
$$+ O(B^3 m^{-1/2} + B^{5/2} m^{-1/4}). \tag{C.27}$$

Telescoping (C.27) for $t = 0, \ldots, T-1$, by $\eta^2 \leq 1/T$ we have

$$\frac{1}{T}\sum_{t=0}^{T-1}\mathbb{E}_{\text{init},\mu}\Big[\big(\widehat{Q}_0(x; W(t)) - \widehat{Q}_0(x; W^*)\big)^2\Big]$$
$$\leq \frac{\mathbb{E}_{\text{init}}\big[\|W(t) - W^*\|_2^2\big] + \sigma_g^2}{T \cdot \big(2\eta(1-\gamma) - 8\eta^2\big)} + O(B^3 m^{-1/2} + B^{5/2} m^{-1/4})$$
$$\leq \frac{B^2 + \sigma_g^2}{\sqrt{T}} \cdot \frac{1}{\sqrt{T} \cdot \big(2\eta(1-\gamma) - 8\eta^2\big)} + O(B^3 m^{-1/2} + B^{5/2} m^{-1/4}), \tag{C.28}$$

where $\eta = \min\{1/\sqrt{T}, (1-\gamma)/8\}$. Note that when $T \geq (8/(1-\gamma))^2$, we have $\eta = 1/\sqrt{T}$ and
$$\sqrt{T} \cdot \left(2\eta(1-\gamma) - 8\eta^2\right) = 2(1-\gamma) - 8/\sqrt{T} \geq 1 - \gamma.$$

Meanwhile, when $T < (8/(1-\gamma))^2$, we have $\eta = (1-\gamma)/8$ and
$$\sqrt{T} \cdot \left(2\eta(1-\gamma) - 8\eta^2\right) = \sqrt{T} \cdot (1-\gamma)^2/8 \geq (1-\gamma)^2/8.$$

Since $|1-\gamma| < 1$, we obtain that for any $T \in \mathbb{N}$,
$$\frac{1}{\sqrt{T} \cdot \left(2\eta(1-\gamma) - 8\eta^2\right)} \leq \frac{8}{(1-\gamma)^2}. \tag{C.29}$$

Similar to (C.24) and (C.26), by combining (C.28) and (C.29) with Lemma 5.1, we obtain
$$\mathbb{E}_{\text{init},\mu}\left[\left(\widehat{Q}_{\text{out}}(x) - \widehat{Q}_0(x; W^*)\right)^2\right] \leq \frac{16(B^2 + \sigma_g^2)}{\sqrt{T} \cdot (1-\gamma)^2} + O(B^3 m^{-1/2} + B^{5/2} m^{-1/4}),$$

which completes the proof of Theorem 4.6. $\qquad\square$

# D   Auxiliary Lemmas

Under Assumption 4.3, we establish the following auxiliary lemmas on the random initialization $W(0)$ and the stationary distribution $\mu$, which plays a key role in quantifying the error of local linearization.

**Lemma D.1.** There exists a constant $c_1 > 0$ such that for any random vector $W$ with $\|W - W(0)\|_2 \leq B$, it holds that
$$\mathbb{E}_{\text{init},\mu}\left[\frac{1}{m}\sum_{r=1}^{m} \mathbb{1}\{|W_r(0)^\top x| \leq \|W_r - W_r(0)\|_2\}\right] \leq c_1 B \cdot m^{-1/2}. \tag{D.1}$$

*Proof.* By Assumption 4.3, we have
$$\mathbb{E}_{\text{init},\mu}\left[\frac{1}{m}\sum_{r=1}^{m} \mathbb{1}\{|W_r(0)^\top x| \leq \|W_r - W_r(0)\|_2\}\right]$$
$$\leq \mathbb{E}_{\text{init}}\left[\frac{1}{m}\sum_{r=1}^{m} c_0 \cdot \|W_r - W_r(0)\|_2 / \|W_r(0)\|_2\right]. \tag{D.2}$$

Applying Hölder's inequality to the right-hand side, we obtain
$$\mathbb{E}_{\text{init},\mu}\left[\frac{1}{m}\sum_{r=1}^{m} \mathbb{1}\{|W_r(0)^\top x| \leq \|W_r - W_r(0)\|_2\}\right]$$
$$\leq c_0/m \cdot \mathbb{E}_{\text{init}}\left[\left(\sum_{r=1}^{m} \|W_r - W_r(0)\|_2^2\right)^{1/2} \cdot \left(\sum_{r=1}^{m} \frac{1}{\|W_r(0)\|_2^2}\right)^{1/2}\right]$$
$$\leq c_0 B \cdot m^{-1/2} \cdot \mathbb{E}_{w \sim N(0, I_d/d)}\left[1/\|w\|_2^2\right]^{1/2}, \tag{D.3}$$

where the second inequality follows from
$$\mathbb{E}_{\text{init}}\left[\left(\sum_{r=1}^{m} \frac{1}{\|W_r(0)\|_2^2}\right)^{1/2}\right] \leq \mathbb{E}_{\text{init}}\left[\sum_{r=1}^{m} \frac{1}{\|W_r(0)\|_2^2}\right]^{1/2} = \sqrt{m} \cdot \mathbb{E}_{w \sim N(0, I_d/d)}\left[1/\|w\|_2^2\right]^{1/2}. \tag{D.4}$$

Setting $c_1 = c_0 \cdot \mathbb{E}_{w \sim N(0, I_d/d)}[1/\|w\|_2^2]^{1/2}$, we complete the proof of Lemma D.1. $\qquad\square$

**Lemma D.2.** There exists a constant $c_2 > 0$ such that for any random vector $W$ with $\|W - W(0)\|_2 \leq B$, it holds that
$$\mathbb{E}_{\text{init}}\left[\mathbb{E}_\mu\left[\widehat{Q}_0(x)^2\right] \cdot \mathbb{E}_\mu\left[\frac{1}{m}\sum_{r=1}^{m} \mathbb{1}\{|W_r(0)^\top x| \leq \|W_r - W_r(0)\|_2\}\right]\right] \leq c_2 B \cdot m^{-1/2}. \tag{D.5}$$

*Proof.* By the definition of $\widehat{Q}_0(x) = \widehat{Q}_0(x; W(0))$ in (4.7), we have

$$\mathbb{E}_\mu\big[\widehat{Q}_0(x)^2\big] = 1/m \cdot \mathbb{E}_\mu\Big[\sum_{r=1}^m \sigma\big(W_r(0)^\top x\big)^2 + \sum_{r \neq s} b_r b_s \sigma\big(W_r(0)^\top x\big)\sigma\big(W_s(0)^\top x\big)\Big].$$

Following the same derivation of (D.2) and (D.3), we have

$$\mathbb{E}_{\text{init}}\Big[\mathbb{E}_\mu\big[\widehat{Q}_0(x)^2\big] \cdot \mathbb{E}_\mu\Big[\frac{1}{m}\sum_{r=1}^m \mathbb{1}\{|W_r(0)^\top x| \leq \|W_r - W_r(0)\|_2\}\Big]\Big]$$

$$\leq \mathbb{E}_{\text{init}}\Big[1/m \cdot \mathbb{E}_\mu\Big[\sum_{r=1}^m \sigma\big(W_r(0)^\top x\big)^2 + \sum_{r \neq s} b_r b_s \sigma\big(W_r(0)^\top x\big)\sigma\big(W_s(0)^\top x\big)\Big]$$

$$\cdot c_0/m \cdot \Big(\sum_{r=1}^m \|W_r - W_r(0)\|_2^2\Big)^{1/2} \cdot \Big(\sum_{r=1}^m \frac{1}{\|W_r(0)\|_2^2}\Big)^{1/2}\Big].$$

Note that $b_r$ and $b_s$ are independent of $W(0)$ and $\mathbb{E}_{\text{init}}[b_r b_s] = 0$. Thus, we obtain

$$\mathbb{E}_{\text{init}}\Big[\mathbb{E}_\mu\big[\widehat{Q}_0(x)^2\big] \cdot \mathbb{E}_\mu\Big[\frac{1}{m}\sum_{r=1}^m \mathbb{1}\{|W_r(0)^\top x| \leq \|W_r - W_r(0)\|_2\}\Big]\Big]$$

$$\leq c_0 B/m^2 \cdot \mathbb{E}_{\text{init}}\Big[\mathbb{E}_\mu\Big[\sum_{r=1}^m \sigma\big(W_r(0)^\top x\big)^2\Big] \cdot \Big(\sum_{r=1}^m \frac{1}{\|W_r(0)\|_2^2}\Big)^{1/2}\Big].$$

By the definition of $\sigma(W_r(0)^\top x)$ and the fact that $\|x\|_2 = 1$, we have

$$\mathbb{E}_\mu\Big[\sum_{r=1}^m \sigma\big(W_r(0)^\top x\big)^2\Big] \leq \sum_{r=1}^m \|W_r(0)\|_2^2.$$

Hence, it holds that

$$\mathbb{E}_{\text{init}}\Big[\mathbb{E}_\mu\big[\widehat{Q}_0(x)^2\big] \cdot \mathbb{E}_\mu\Big[\frac{1}{m}\sum_{r=1}^m \mathbb{1}\{|W_r(0)^\top x| \leq \|W_r - W_r(0)\|_2\}\Big]\Big]$$

$$\leq c_0 B/m^2 \cdot \mathbb{E}_{\text{init}}\Big[\Big(\sum_{r=1}^m \|W_r(0)\|_2^2\Big) \cdot \Big(\sum_{r=1}^m \frac{1}{\|W_r(0)\|_2^2}\Big)^{1/2}\Big]$$

$$\leq c_0 B/m^2 \cdot \mathbb{E}_{\text{init}}\Big[\Big(\sum_{r=1}^m \|W_r(0)\|_2^2\Big)^2\Big]^{1/2} \cdot \mathbb{E}_{\text{init}}\Big[\sum_{r=1}^m \frac{1}{\|W_r(0)\|_2^2}\Big]^{1/2}. \qquad (D.6)$$

By (D.4) and the fact that

$$\mathbb{E}_{\text{init}}\Big[\Big(\sum_{r=1}^m \|W_r(0)\|_2^2\Big)^2\Big] = m \cdot \mathbb{E}_{w \sim N(0, I_d/d)}\big[\|w\|_2^4\big] + m(m-1) \cdot \mathbb{E}_{w \sim N(0, I_d/d)}\big[\|w\|_2^2\big]^2 = O(m^2),$$

the right-hand side of (D.6) is $O(Bm^{-1/2})$. Setting

$$c_2 = c_0 \cdot \Big(\mathbb{E}_{w \sim N(0, I_d/d)}\big[\|w\|_2^4\big] + \mathbb{E}_{w \sim N(0, I_d/d)}\big[\|w\|_2^2\big]^2\Big)^{1/2} \cdot \mathbb{E}_{w \sim N(0, I_d/d)}\big[1/\|w\|_2^2\big]^{1/2},$$

we complete the proof of Lemma D.2. $\qquad\square$