[Reviews · NeurIPS 2019]

Reviewer 1



This paper studied the non-asymptotic convergence rate of the TD algorithm under overparameterized neural network approximation. For such nonlinear function approximation, TD algorithm may not even converge. The paper exploits the fact that the overparameterized neural network has implicit local linearization, which guarantees the convergence of the TD algorithm to global convergence. The paper also generalizes the analysis to Q-learning and policy gradient (in appendix). This is the first study that made connection to recent breakthrough analysis of overparameterized neural networks, and leveraged the property for establishing the convergence rate for reinforcement learning algorithms. The contribution is solid. The presentation of the paper is clear and easy to follow. I have the following question for the authors to clarify. The neural TD algorithm adopts the average of the parameter matrix W as the output. It allows the constant stepsize, but requires preset the number of iteration T. Is there any specific reason to adopt such an algorithm? What happens for the simple semi-SGD output without taking the average? This may require diminishing stepsize to converge or constant stepsize to converge to a neighborhood of the global minimum. But in terms of the analysis, will the current approach be applicable?

Reviewer 2



Originality: ========= The paper relies on recent results on the implicit local linearization effect of overparametrized neural networks in the context of supervised learning, and on recent nonasymptotic analysis of Linear TD and Linear Q-learning. Perhaps the main insight is the relationship between the explicit linearization of Linear TD and the implicit linearization of overparametrized neural TD. Related work is properly referenced. Quality: ========= The paper seems to be technically sound (although I have just skimmed over the proofs). The convergence of the three algorithms, namely Neural TD, Neural Q-learning, and Neural Soft Q-learning constitute a complete piece of work. The authors explain the assumptions for the analysis, and discuss their generality and how they relate to previous literature. The only confusion on my side is that the authors talk about 2 layer neural networks, which made me think about a deep architecture with 2 hidden layers. However, Eq. (3.2) seems to define a single hidden layer with linear output. Clarity ========= The paper is clearly well written and well organised. Significance ========= The fact that they have been able to use the same approach to study nonasymptotic convergence of three different Bellman operators is promising. In addition, it is worth to remark that similar ideas to those presented here have been used to study a control algorithm in another submission titled "Neural Proximal Policy Optimization Attains Optimal Policy (5585)," potentially with overlapping authors. However, such submission is different enough from this one, which highlights the usefulness of the ideas presented here. Although the authors assume a simple neural network architecture, I imagine that similar linearization effects can be explored in more complex architectures.

Reviewer 3



The paper consists of solid enough efforts to analyze the convergence rate. While I am convinced by the results, I cannot verify the details of the proof. I have some questions regarding the theory, e.g. I had a difficulty of finding the notion of "one-point monotonicity" in the paper, which probably should be included for self-containing. The assumption 4.3, 5.2, incorporate some constants c1, c3, \mu, however, at least in the main paper, there is no clear explanation of how these constants relate to the convergence rate. The paper involves a two-layer neural network, but still, it seems to be like a linearly parameterized model, as well as the rate, if all these assumptions are to apply. The implication of theorem 4.4, 4.6 has not made clear how the width of neural network effects the rate, and it seems to be hidden in the big-O notation, so it is hard for me to value the importance of overparameterization, or explicitly, the number of nodes in this two-layer neural network, comparing to number of iterations.

[Author Response · NeurIPS 2019]

We appreciate the valuable comments and positive feedback from the reviewers. We will carefully revise the paper
accordingly to incorporate the comments.

**Reviewer #1:** (**Stepsize and preset $T$.**) Following the current analysis, for a general stepsize $\eta_t$, the convergence of
stochastic update requires $(\sum_{t=1}^{T} \eta_t^2)/(T \cdot \min_{t \leq T} \eta_t) \to 0$ and $T \cdot \min_{t \leq T} \eta_t \to \infty$ as $T \to \infty$ to handle the
variance of stochastic semigradient. Thus, a diminishing adaptive stepsize such as $\eta_t = 1/\sqrt{t}$ would also work, but the
convergence rate would then become $O(\log T/\sqrt{T})$, which is slightly slower than the $O(1/\sqrt{T})$ rate in our paper. For
the same reason, an absolute constant stepsize does not guarantee convergence, since it fails to satisfy the first
requirement. In view of the two requirements, we use the stepsize $\eta_t = 1/\sqrt{T}$ to obtain the fastest rate $O(1/\sqrt{T})$. We
will add a corresponding discussion in the revision.

(**Average of iterates.**) In the current analysis, the convergence rate is implied by the upper bound of a telescope sum
(line 618 of the full paper). Without averaging the iterates, no convergence rate is available. Although the iterates using
population semigradient would still converge, stochastic semigradient might cause divergence. Such a situation is
analogous to convex optimization without strong convexity, where averaging the iterates is necessary [1].

**Reviewer #2:** (**Two-layer neural network.**) In this paper we consider neural network with one hidden layer. It is
called a two-layer neural network following the recent line of work (e.g., [2]), since there is also a linear output layer.
We recognize the potential confusion in terminology and will explicitly clarify that we mean a two-layer net with a
single hidden layer.

(**Motivation for choosing the architecture.**) Such a shallow structure helps to characterize the learning dynamics and
illustrate the connection to linear model with random features. With one hidden layer, it is already quite challenging to
analyze the effects of using overparametrized neural networks for function approximation in RL.

(**Generalization to more complex networks.**) The results can be readily generalized to deep neural networks
(multiple hidden layers with width $m$) given the activation function is sufficiently smooth (e.g., sigmoid activation) and
each layer is coupled with a suitable scaling factor. However, the ReLU activation used in this paper does not directly
satisfy the smoothness requirement and therefore requires more delicate analysis.

(**MSPBE with oblique projections.**) Thanks for bringing up the oblique projection view. We will add a corresponding
discussion in the revision. In the oblique projection paper, the difference between temporal difference-based and
Bellman residual-based approaches arises due to the limited representation power of finite-dimensional linear function
approximation. In comparison, overparametrized neural networks represent a larger infinite-dimensional function class,
which alleviates the issues caused by limited representation power and therefore bridges the gap between the two
approaches. In particular, Proposition 4.7 shows that neural TD attains the global minimum of MSBE (without the
projection in MSPBE) under slightly stronger conditions.

(**State assumption.**) Our proof only relies on the fact that $x$ is bounded, while the unit-norm assumption is used to
simplify the presentation. An alternative view of this assumption is that the neural network has an additional (fixed)
input layer that projects or embeds the "raw input" $(s, a) \in \mathcal{S} \times \mathcal{A}$ to the unit sphere.

(**Reward assumption.**) Thanks for pointing this out. Coercive reward indeed requires more delicate analysis and is
beyond the scope of this paper. We will revise the "without loss of generality" claim in the revision.

(**Function class $\mathcal{F}_{B,\infty} - \hat{Q}(\cdot; W(0))$.**) For any function class $\mathcal{F}$ and function $f'$, the function class $\mathcal{F} - f'$ is defined
as $\{g = f - f' : f \in \mathcal{F}\}$. We will clarify this notation in the revision.

(**Minor comments.**) Thanks for pointing out the issues on notation and clarity. We will fix them in the revision.

**Reviewer #3:** (**One-point monotonicity.**) See line 591 of the full paper (deferred due to space limit) for more details
on the notion of one-point monotonicity. We will move this to the main text in the revision as it is an important concept
for this paper. Thank you for pointing this out.

(**Constants $c_1$, $c_3$, $\nu$.**) The exact polynomial dependency on $c_1$ and $c_3$ in the convergence rate is quantified in Lemma
A.2 and the proof of Lemma E.2 (line 802) of the full paper (deferred due to space limit), which is omitted in big-O's
when the lemmas are invoked. Meanwhile, the dependency on $\nu$ is quantified in the proof of Theorem 5.3 (inequality
(E.23) of the full paper) and is of order $O(1/\nu)$. We will move the dependencies to the main text in the next version.

(**How width affects rate.**) The effect of overparametrization is explicitly quantified in Theorems 4.4, 4.6, and 5.3 by
the terms that decay with $m$, which denotes the width of the neural network. Roughly speaking, the convergence rate
takes the form of $1/\sqrt{T} + 1/\sqrt{m}$. As $m \to \infty$ (or at least $m = \Omega(T)$), the rate reduces to $1/\sqrt{T}$, where $T$ is the
number of iterations. In other words, the "error of implicit linearization" diminishes as the neural network has more
parameters. We will include a discussion of how width affects the convergence rate in the next revision.

[1] Bubeck, S. (2015). Convex optimization: Algorithms and complexity. Foundations and Trends in Machine
Learning, 8 231–357.

[2] Arora, S., Du, S. S., Hu, W., Li, Z. and Wang, R. (2019). Fine-grained analysis of optimization and generalization
for overparameterized two-layer neural networks. arXiv preprint arXiv:1901.08584.


[Meta-Review · NeurIPS 2019]

The reviewers seem to be in agreement that this is above the acceptance threshold.